# A minimally sufficient model for rib proximal-distal patterning based on genetic analysis and agent-based simulations

Jennifer L Fogel[1†], Daniel L Lakeland[2†], In Kyoung Mah[1], Francesca V Mariani[1]*

[1]Eli and Edythe Broad Center for Regenerative Medicine and Stem Cell Research, University of Southern California, Los Angeles, United States; [2]Lakeland Applied Sciences LLC, Altadena, United States

**Abstract** For decades, the mechanism of skeletal patterning along a proximal-distal axis has been an area of intense inquiry. Here, we examine the development of the ribs, simple structures that in most terrestrial vertebrates consist of two skeletal elements—a proximal bone and a distal cartilage portion. While the ribs have been shown to arise from the somites, little is known about how the two segments are specified. During our examination of genetically modified mice, we discovered a series of progressively worsening phenotypes that could not be easily explained. Here, we combine genetic analysis of rib development with agent-based simulations to conclude that proximal-distal patterning and outgrowth could occur based on simple rules. In our model, specification occurs during somite stages due to varying Hedgehog protein levels, while later expansion refines the pattern. This framework is broadly applicable for understanding the mechanisms of skeletal patterning along a proximal-distal axis.

DOI: https://doi.org/10.7554/eLife.29144.001

*For correspondence:
fmariani@usc.edu

†These authors contributed equally to this work

Competing interests: The authors declare that no competing interests exist.

## Introduction

During evolution, a number of changes in vertebrate body plan allowed terrestrial tetrapod species to thrive on land and take advantage of new habitats. For example, in contrast to the open rib cages of aquatic species, the thoracic cage became an enclosed chamber that could support the weight of the body and facilitate lung ventilation (*Janis and Keller, 2001*). Current tetrapod species typically have ribs that are subdivided into two segments, a proximal endochondral bony segment connected to the vertebrae, and a distal permanent cartilage segment that articulates with the sternum (*Figure 1A,B*). The rib bones support the body wall and protect the internal organs; the costal cartilage maintains thoracic elasticity, allowing respiration while still enclosing the thoracic cage. Although clues from the fossil record are beginning to reveal when the enclosed rib cage arose during evolution (*Daeschler et al., 2006*; *Pierce et al., 2013*), little is known about what changes occurred during embryogenesis to extend the ribs around the body and to connect the ribs to the sternum via a costal cartilage element (*Brainerd and Brainerd, 2015*). Here, using genetic and computational approaches, we generate a plausible model for how two rib segments form during development.

Lineage-tracing studies indicate that the sternum and ribs have different developmental origins. The sternum, like the appendicular skeleton, arises from the lateral plate mesoderm (*Cohn et al., 1997*; *Bickley and Logan, 2014*), while the ribs and vertebrae arise from the somites (reviewed in [*Brent and Tabin, 2002*]). Studies using chicken-quail chimera grafts have shown that the thoracic somites contribute to all portions of the ribs (*Huang et al., 1994*), with a the medial somite

**eLife digest** During animal development, the ribs grow from the back of the embryo around towards the chest. In fish, these bones simply terminate. Yet in land animals, cartilage forms at the end of the rib where it connects to the breastbone, or sternum. This encloses the chest cavity.

Fogel, Lakeland et al. have now asked how the progenitor cells that develop into the ribs form these two skeletal elements – the bone element and the cartilage element – in land animals. Their approach involved genetic analysis in mice and a simple computing model. It revealed that two elements could form if the progenitor cells decide which element they will belong to based on the concentration of the diffusible protein called Hedgehog. This protein controls many aspects of animal development, and higher concentrations seem to bias the cells in a developing rib toward belonging to the bone element. Fogel, Lakeland et al. propose that this decision is locked-in early, before the rib grows outward and becomes more refined. An analysis using this simple model reproduces all the basic observations seen in the experiments with mice. The model also explains how processes like cell division and cell death control the growth of developing skeletal elements.

These modeling techniques can be applied to many fields within biology, including research into the causes of birth defects, the mechanisms of tissue repair, and the evolution of skeletal diversity. An advantage to this modeling technique is that it uses only the information in each cell's local environment to make decisions.

DOI: https://doi.org/10.7554/eLife.29144.002

contributing to the proximal ribs while lateral somite contributes to the distal ribs (*Olivera-Martinez et al., 2000*). These results suggest that the proximal and distal progenitor populations of the rib are distinct at early somite stages rather than being intermixed. As the whole somite matures, it separates into distinct dorsal (dermomyotome and myotome) and ventral (sclerotome) compartments (*Figure 1C*). Initially, there was some debate on the precise embryological origin of the ribs within the somite (*Kato and Aoyama, 1998*; *Huang et al., 2000*). However, using retroviral lineage labeling which avoids the challenges of transplantation experiments, both the proximal and distal segments of the rib were shown to arise from the sclerotome compartment (*Evans, 2003*). It has been still unclear though, how the sclerotome becomes patterned along the proximal-distal axis.

Through studies particularly of *Drosophila* wing/leg disc and of vertebrate limb development over the past decades, several patterning models have been conceived to explain how proximal-distal, dorsal-ventral, and anterior-posterior pattern arises (*Briscoe and Small, 2015*). For example, compartments could become specified based on: (1) the presence of cellular determinants, (2) the concentration of a morphogen, (3) the duration of exposure to a signaling molecule, and/or (4) the action of local relay or mutual inhibition signaling. Specification could gradually emerge over the course of organogenesis or via a biphasic process with specification occurring early in a small population of cells followed later by expansion into compartments (recently reviewed in [*Zhu and Mackem, 2017*]). In this study, we first use genetically modified mice in which the Hedgehog (Hh) and apoptosis pathway is disrupted to provide clues for how two rib segments are patterned and grow. Our experiments produced unexpected results which led us to seek an explanation using Agent-Based Modeling, a simulation method based on a cell's ability to make decisions in response to stimuli. We designed a set of simple rules that could produce a wide variety of potential phenotypes which then motivated the collection of further biological measurements. Using a refined model, we were then able to conclude that complex patterning and growth can emerge through a set of simple rules and biologically supported parameters. Furthermore, our model does not require individual cells to have necessarily received any positional information prior to Hh expression. Finally, we find that our model is essentially biphasic, with early events that define the size and fate of the progenitor populations and later events that expand the already specified population.

Studying rib formation during embryogenesis, and in particular, determining how the segments of the rib cage become distinct, provides a relatively simple test-case for questions regarding skeletal patterning, while also giving clues as to the evolutionary origin of the enclosed rib cage. In addition, our studies may aid in understanding the etiology of congenital abnormalities of the rib cage (*Blanco et al., 2011*). The use of agent-based modeling provides insight into how simple decision

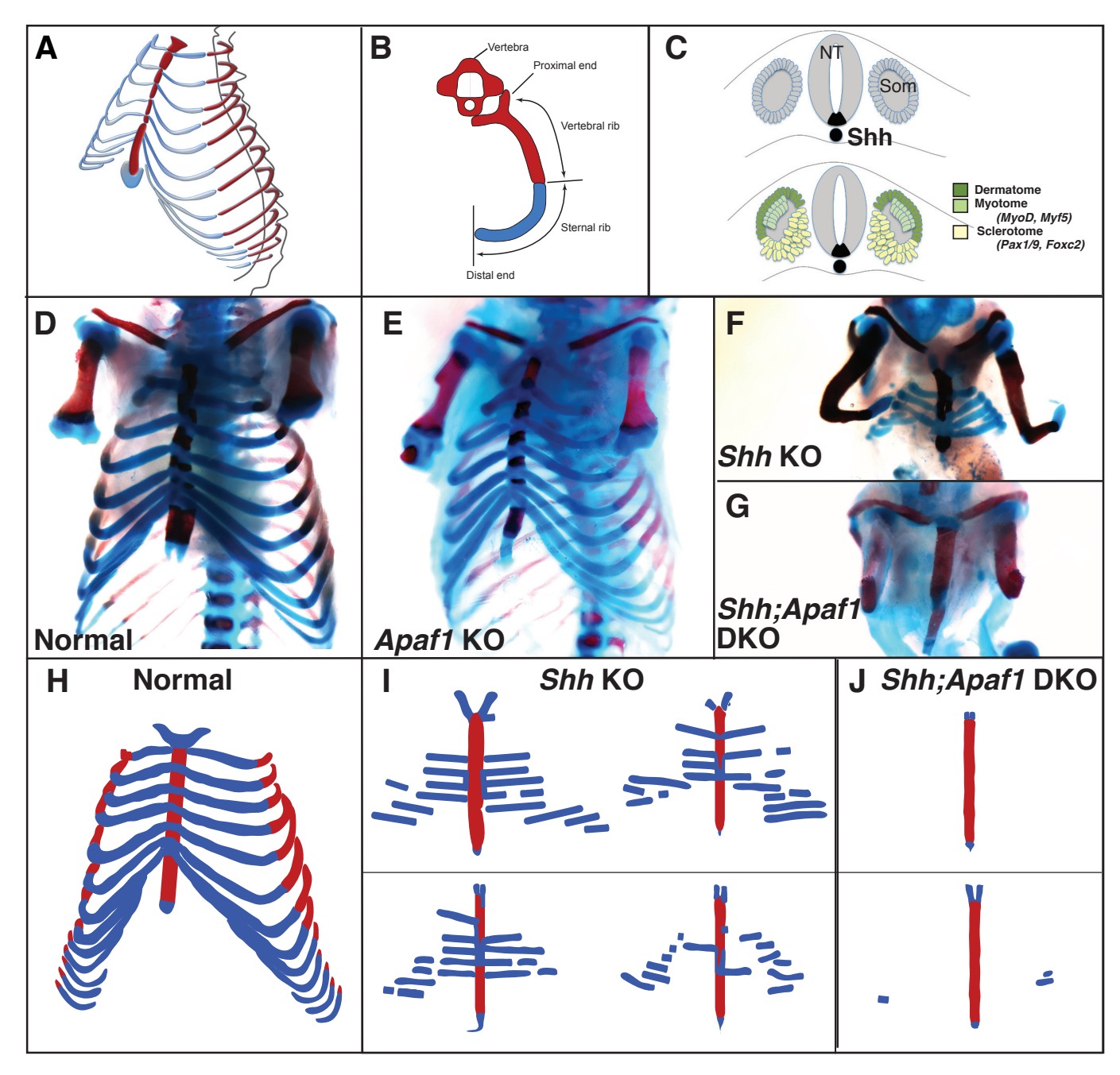

**Figure 1.** Rib skeletal development is compromised in *Shh* null animals. (**A**) Frontal 1/3 view of the thoracic cage depicting the orientation of the proximal and distal ribs. Mice have 13 pairs of ribs. (**B**) Schematic of a vertebra and rib, transverse view. Red represents bone including the proximal/vertebral rib and blue represents the cartilaginous distal/sternal rib. (**C**) The somite (Som), neural tube (NT), and notochord diagramed in cross-section. The dermatome and myotome (dark and light green) gives rise to the dermis and muscles while the sclerotome (yellow) gives rise to the vertebrae and ribs. Markers for these compartments are indicated. The location of *Shh*-expressing cells is indicated in black. (**D–G**) Alcian blue and alizarin red skeletal staining of the rib cage: (**D**) Normal mouse skeletal development at E18.5 (lower arms removed). (**E**) Without *Apaf1*, the axial skeleton develops normally (n = 12/12, lower arms removed). (**F**) Without *Shh*, embryos develop without vertebrae and the proximal portion of the ribs. Distal ribs form but are not patterned correctly (n = 17/17). (**G**) Double knock out (DKO) of both *Shh* and *Apaf1* results in a more severe phenotype. DKO neonates develop without vertebrae, proximal *and* distal ribs (n = 7/7). The sternum is still present and ossifies on schedule. (**H–J**) Schematics representing skeletal preparations of normal (**H**) and null neonates. (**I**) The loss of the proximal ribs is consistent amongst all *Shh* KO neonates, however, the disrupted pattern of the distal ribs vary. (**J**) Occasionally *Shh;Apaf1* DKO neonates have cartilage nodules laterally (presumably at the chondro-costal joint, n = 1/7).

*Figure 1 continued on next page*

*Figure 1 continued*

DOI: https://doi.org/10.7554/eLife.29144.003

making at the cellular level could lead to the emergence of multi-component structures during skeletal development.

## Results

### Impact of blocking cell death in *Shh* null embryos

Previous studies have demonstrated the importance of Hedgehog (Hh) signaling for sclerotome induction and specification. Overexpression of Sonic hedgehog (Shh) can produce ectopic sclerotome at the expense of dermomyotome. Furthermore, in the absence of *Shh*, *Pax1* expression is reduced and later lost; vertebrae and the proximal rib segments then fail to form while the distal cartilaginous portions of the ribs are still present, although abnormal (*Fan and Tessier-Lavigne, 1994*; *Chiang et al., 1996*; *Marcelle et al., 1999*). Hh signaling is also well-known to be important for promoting cell proliferation, growth, and survival (*Charrier et al., 2001*; *Thibert et al., 2003*), and in the absence of Shh or Shh-producing cells in the floor plate and notochord, the apoptosis in the somite is greatly increased (*Teillet et al., 1998*). Thus, in addition to a role for Hh signaing in somite/sclerotome induction, Hh signaling also protects somite cells from undergoing programmed cell death. To distinguish between these two roles, we decided to block cell death in *Shh* null animals. In previous studies, where loss of *Shh* results in high cell death in the developing heart and face, removing the function of Apoptotic protease-activating factor 1 (Apaf1), a central player in the mitochondrial pathway of programmed cell death, could block cell death and rescue the phenotype (*Aiyer et al., 2005*; *Long et al., 2013*). Perhaps similarly, removing *Apaf1* on a *Shh* null background would inhibit somite cell death and rescue the thoracic phenotype. We therefore generated genetically modified mouse lines that lacked *Shh*, *Apaf1*, or both.

Apaf1 is required for the Cytochrome c and ATP-dependent activation of Caspase9 which leads to the subsequent activation of Caspase3, followed by the initiation of nuclear breakdown and proteolysis (*Zou et al., 1997*). Embryos carrying null alleles for *Apaf1* (*Apaf1* KO) have vastly decreased cell death in the nervous system and exhibit disruptions of the head and face skeleton likely due to a grossly overgrown CNS. In addition, interdigital death in the autopod is delayed (*Cecconi et al., 1998*; *Yoshida et al., 1998*). Although, the embryos develop to term, they typically die at birth and have not been examined during skeletal development. Therefore, we investigated the skeleton of *Apaf1* KO embryos and found that patterning of the axial and appendicular portions is largely normal and that cartilage and bone development proceeds on schedule, indicating that *Apaf1* is not required for normal skeletal development (*Figure 1D,E*).

As previously observed, *Shh* KO animals exhibit a failure to form the vertebral column and pronounced rib cage defects (*Figure 1F*) (*Fan and Tessier-Lavigne, 1994*; *Chiang et al., 1996*). Absence of the proximal ribs was consistently observed amongst all *Shh* KO embryos stained for bone and cartilage (E15-18). In contrast, the pattern of the remaining rib segments varied (*Figure 1I*). These cartilage portions were distally located and never mineralized suggesting that they represented the distal costal cartilages. These segments were not entirely normal as they were discontinuous, reduced in number (~7–8 instead of 13), not properly articulated with the sternum, and positioned at abnormal angles. The clavicle and sternum were present, although the sternum was sometimes not completely fused and had missing or disorganized segments; however, it did undergo ossification on schedule. In a few cases, small condensations could be observed laterally, possibly near the chondro-costal joint (*Figure 1F*).

We then created animals double null for *Shh* and *Apaf1* (*Shh;Apaf1* DKOs) and analyzed the skeletal pattern. To our surprise, instead of observing a rescue of the *Shh* KO phenotype, *Shh;Apaf1* DKO animals exhibited an even more severe skeletal phenotype. *Shh;Apaf1* DKO embryos did display features of the *Shh* single KO (no vertebrae and proximal ribs). However, in addition to these defects, the distal portion of the ribs was now missing as evidenced by the lack of alcian blue staining in the body wall (*Figure 1G*). Rarely, a few small pieces of cartilage were present at the lateral margin, near the expected location of the chondro-costal joint (*Figure 1J*).

## Similar synergistic failure in axial skeleton formation in the absence of *Caspase3*

Apaf1 has been shown to have other roles in addition to regulating the programmed cell death pathway (*Zermati et al., 2007*; *Ferraro et al., 2011*). Thus, to determine if the observed effects were specific to Apaf1, we carried out mouse crosses utilizing a *Caspase3* null allele (*Casp3* KO). Caspase3 is an executioner caspase, that cleaves key structural proteins leading to DNA fragmentation and membrane blebbing (*Fuchs and Steller, 2011*). As with loss of *Apaf1*, the axial skeleton of *Casp3* KO animals was grossly normal (*Figure 2A,B*). *Shh;Casp3* DKO embryos exhibited a complete absence of all vertebrae, vertebral ribs and sternal ribs as was observed in the *Shh;Apaf1* DKO embryos (compare *Figure 1G* and *Figure 2D*). These results suggest that the observed phenotypes when *Apaf1* is removed are not due to a specific non-canonical function but rather due to a general abrogation of the programmed cell death pathway.

## Embryos null for *Apaf1 or Casp3* exhibit a reduction in cell death

To determine if the absence of programmed cell death genes was indeed preventing cells from dying, TUNEL and/or LysoTracker assays were performed (*Fogel et al., 2012*). In control embryos, evidence of normal cell death can be seen beginning as early as E9.0 as distinct periodic stripes of staining in the somites, extending to E10.5 and waning by E11.5 (*Figure 2E,E', I*) (*Sanders, 1997*; *Teillet et al., 1998*). The absence of *Apaf1*, however, results in a dramatic reduction in cell death throughout the embryo, and notably absence of cell death in the somites (*Figure 2F,F'*). In contrast, *Shh* KO embryos exhibit high cell death in the somites — with high LysoTracker-positive staining in the ventral sclerotome domain (*Figure 2G,G'* and inset, marked with brackets). Interestingly, the absence of *Apaf1* in *Shh* null embryos (*Shh;Apaf1* DKOs) results in a vast reduction in LysoTracker-positive cells (*Figure 2H,H'*). Like *Apaf1* KO embryos, *Casp3* KO embryos display a dramatic reduction in cell death compared to the normal pattern (*Figure 2I,J*). Similarly, *Shh;Casp3* DKO embryos (*Figure 2L*), also exhibit a dramatic reduction in cell death compared to the *Shh* KO pattern. Thus, the absence of either *Apaf1* or *Casp3* results in an inhibition of programmed cell death even in *Shh* null animals. However, it seems counterintuitive that a reduction in cell death could lead to a more severe skeletal phenotype.

## *Shh* expressed in the floor plate and notochord is necessary for rib development.

Previous studies have demonstrated that *Shh* expression in the notochord and floor plate is essential for ventral neural tube and somite specification (*Varjosalo and Taipale, 2008*). However, *Shh* is also expressed in other developing tissues that could affect somite patterning and rib formation. For example, RNA in situ hybridization reveals that *Shh* is also expressed in the dorsal root ganglia, ventral neural tube, developing lungs, as well as the developing ribs themselves (*Figure 3A,B*). To determine if *Shh* specifically from the notochord and floor plate is required to obtain the observed phenotypes, *Shh* hypomorph embryos were created utilizing a tamoxifen inducible *Foxa2*-CRE conditional knock-out approach (*Park et al., 2008*). Using this system, the temporal influence of Shh on rib development could also be analyzed. Administration of tamoxifen at E8.0 did not alter *Ptch1* expression, a readout of *Shh* signaling, in E9.0 somites indicating that the activity of Shh prior to cre-mediated deletion (likely sometime between E8.0-E10) was sufficient for *Ptch1* expression. As a result, these embryos had no skeletal defects in the thoracic cage; later injections also failed to generate skeletal defects. However, administration of tamoxifen at the same doses at E7.0 caused a discontinuity in *Shh* and *Ptch1* expression at E8.0 (see *Figure 3—figure supplement 1*). Subsequently, embryos developed with a range of *Shh* KO hypomorphic phenotypes (*Figure 3*). Importantly, the most severe phenotypes were very similar to *Shh* KO animals and lacked the vertebrae and proximal ribs (*Figure 3C,F*). The distal-most sternal ribs were present but mis-patterned, although less severely than the *Shh* KO animals. Furthermore, being additionally null for *Casp3* resulted in failed distal rib development (*Figure 3D*). Among the hypomorphic *Shh* KO animals, less severely affected animals lacked vertebrae and the proximal half of the vertebral rib, while the least affected only had abnormal vertebrae and were missing just the most proximal ends of the proximal rib (*Figure 3F–H*). Thus, these experiments indicated that Shh signaling from the notochord and floor plate is required for normal thoracic skeletal development at stages prior to rib outgrowth and potentially even

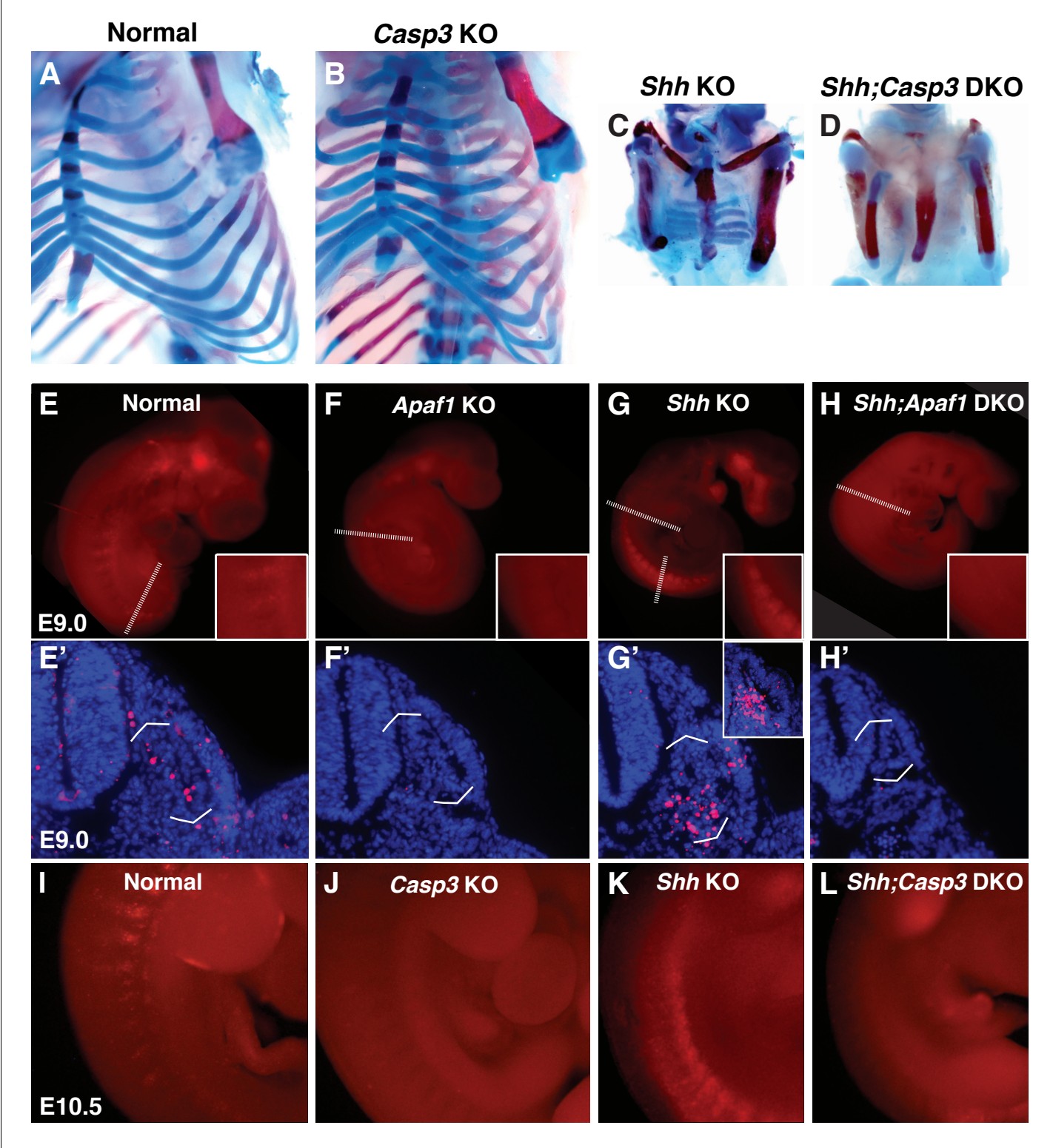

**Figure 2.** Programmed cell death is decreased in the absence of *Casp3* and *Apaf1*. (A–D) Alcian blue and alizarin red staining, E18.5 neonates. (A, B) Normal and *Casp3* KO embryos are indistinguishable. (C) *Shh* KO embryos develop without vertebrae and proximal ribs and as before, distal ribs are present but mispatterned. (D) *Shh;Casp3* DKO embryos exhibit a similar phenotype to *Shh;Apaf1* DKO embryos— a complete loss of all vertebrae, proximal, and distal ribs. (E–L) Embryos stained with LysoTracker (red) are shown in whole-mount and in section (blue, DAPI). Dashed lines indicate the location of the sections in the panels below. (E, E') Normal embryos have LysoTracker staining in the roof plate, limbs, and in the developing somites.
*Figure 2 continued on next page*

*Figure 2 continued*

(F, F') Throughout *Apaf1* KO embryos, including the somites, LysoTracker staining is dramatically reduced. (G, G') *Shh* null embryos have increased LysoTracker staining in the somites. At the thoracic level, staining is found in sclerotomal cells emerging from the ventral medial somite. At a more caudal level (inset), abundant staining is present in the sclerotome epithelium. (H, H'). *Shh;Apaf1* DKO embryos have little to no LysoTracker staining throughout. (I–L) Embryos from the *Shh-Caspase3* cross had similar LysoTracker staining patterns. (I) Normal embryos. (J) *Casp3* KO embryos displayed a dramatic reduction in LysoTracker staining. K. *Shh* KO embryos have abundant staining in the somites. (L) Reduced LysoTracker staining. (similar to *Shh;Apaf1* DKO) was also observed in *Shh;Casp3* DKO null embryos. Phenotypes were consistently observed in at least 3/3 animals of each genotype.

DOI: https://doi.org/10.7554/eLife.29144.004

earlier. In addition, a comparison of the milder phenotypes vs. the more severe phenotypes suggests that high levels of Hedgehog signaling are required for normal proximal development and lower levels for distal development.

## Somite patterning does not differ between the *Shh* KO and *Shh;Apaf1* DKO embryos

It is well-established that somite patterning is specified by instructive signals from the surrounding tissues (*Brent and Tabin, 2002*) with Shh from the midline being required for proper sclerotome patterning (*Fan and Tessier-Lavigne, 1994*; *Fan et al., 1995*; *Furumoto et al., 1999*; *Marcelle et al., 1999*). Thus, one possibility is that the more severe skeletal phenotypes in *Shh;Apaf1* DKO animals could be due to defects in somite patterning more profound than observed in *Shh* KO animals. We first found that compared to *Shh* KO embryos, *Shh;Apaf1* DKO embryos were smaller (compare panels 2G', H') and consistently delayed at E9-E12, (~3 fewer somites than average for that litter; n > 6 litters) suggesting the *Apaf1* plays a role in embryo size in the absence of *Shh*. Thus, we decided to compare patterning without size as a variable, by carefully stage-matching embryos by somite count and assessing for myotome and sclerotome patterning by RNA in situ hybridization. Defects in myotome development (which could secondarily impact sclerotome) could account for the more severe *Shh;Apaf1* DKO phenotypes, however, this was not the case (see *Figure 3—figure supplement 2*). Or, sclerotome might never specified in *Shh;Apaf1* DKO animals leading to the absence of both proximal and distal rib elements. However, while the expression domain of sclerotome markers *Pax1* and *FoxC2* (*Furumoto et al., 1999*; *Peters et al., 1999*; *Rodrigo et al., 2003*) are reduced in both *Shh* KO and *Shh;Apaf1* DKO embryos, the expression profiles were similar (*Figure 3—figure supplement 2*). Thus, an alteration in early somite patterning does not readily explain the difference in the final skeletal phenotype between *Shh* KO and *Shh;Apaf1* DKO animals.

## Progenitor cells are specified but are unable to differentiate

We next determined if it is the failure of this remaining sclerotome compartment to undergo differentiation into cartilage that distinguishes *Shh* KO from *Shh;Apaf1* DKO animals (schematic in *Figure 4A*). The differentiation of cartilage involves the specification of mesenchymal cells to a cartilage fate as evidenced by the upregulation of *Sox9*, a master regulator of the cartilage pathway (*Akiyama et al., 2002*). Cells destined to become cartilage then form aggregates which subsequently undergo compaction and condensation to form tight clusters of cells. These cells then begin to produce Aggrecan, Type II Collagen, and a specific matrix rich in acid polysaccharides detectable by alcian blue staining. Finally, the chondrogenic cells mature, slow their production in matrix, and become hypertrophic (reviewed in [*DeLise et al., 2000*]). We first determined if chondroprogenitors are ever specified in *Shh;Apaf1* DKO embryos by examining *Sox9* expression at E12.0. At this stage, *Sox9* expression is observed in control embryos extending from the vertebrae laterally approximately halfway around the chest (*Figure 4B*) and in *Shh* KO embryos in a thinner distally located reduced domain (likely representing the precursors of the distal-most sternal ribs) (*Figure 4C*). Interestingly, *Sox9*-expressing cells are indeed present in *Shh;Apaf1* DKO embryos, however, in an even thinner and shorter domain (*Figure 4D*) indicating that cartilage specification occurs in *Shh;Apaf1* DKO embryos. We next determined if chondrogenic aggregates and condensations expressing *Sox9* and *Agc* could be observed in cross-section. Aggregations and tight condensations could be seen in normal and *Shh* KO embryos (*Figure 4E,F;H,I*). However, in *Shh;Apaf1* DKO embryos, while some aggregation could be observed, distinct condensations were not evident (*Figure 5G,J*). Further differentiation of cartilage involves the expression of *Col2a1*, the continued expression of *Sox9*, and

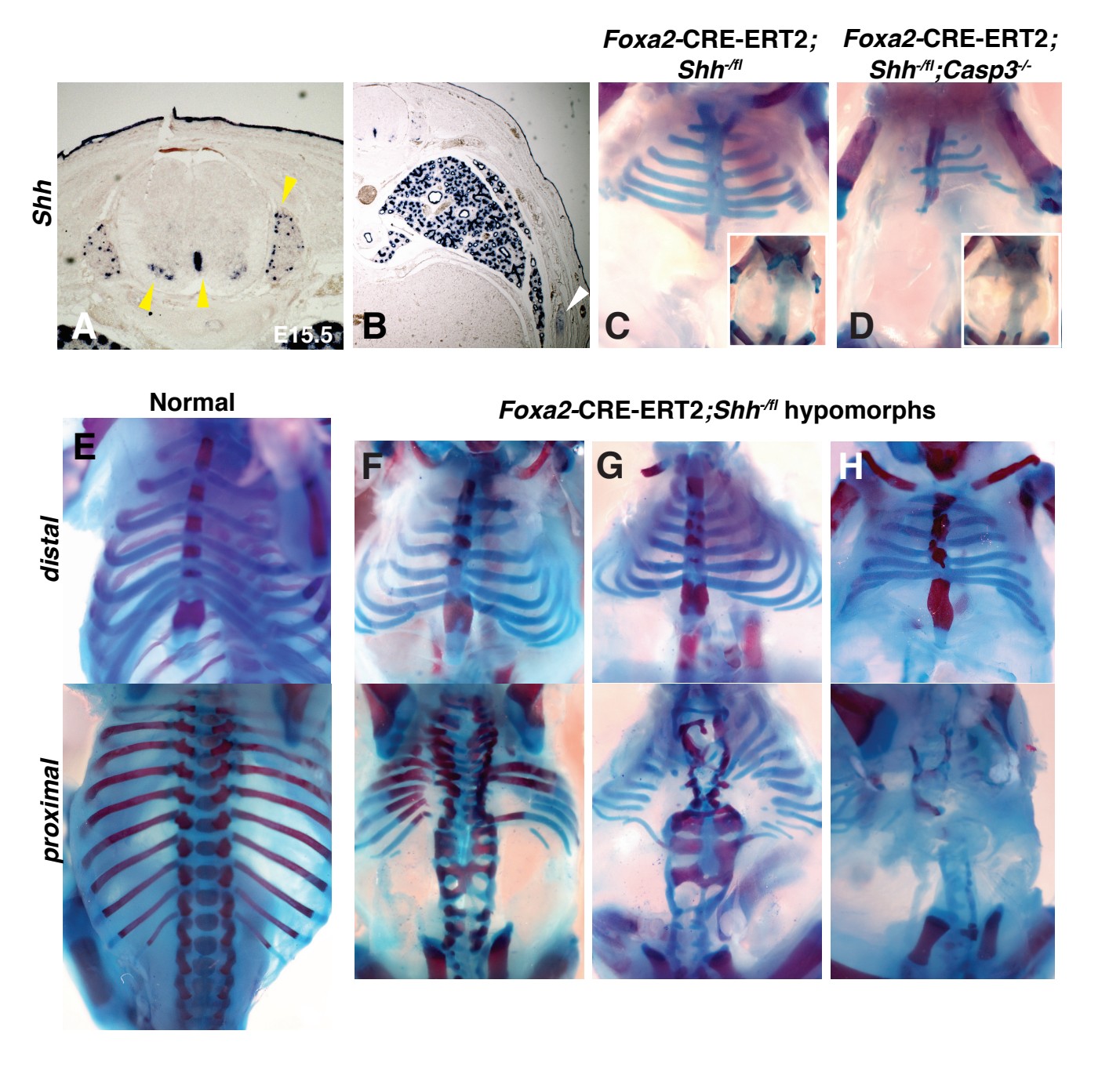

**Figure 3.** Conditional removal of *Shh* shows an early role for Hh signaling in rib development. (A, B) *Shh* is expressed not only in the floor plate but also in the dorsal root ganglia, ventral neural tube, developing lungs (yellow arrowheads), as well as the developing ribs themselves (white arrowhead). (C–H) Alcian blue and alizarin red staining of neonates in which an inducible *Foxa2-CRE-ERT2* was used to ablate *Shh* only in the notochord and floor plate. (C) An E18.5 *Foxa2-CRE-ERT2;Shh⁻/fl* neonate after a Tamoxifen injection at E7.0 exhibits a similar phenotype to *Shh* null animals. Neonates develop without vertebrae and proximal ribs (inset – dorsal view). Distal ribs develop with mild segmental defects. (D) The additional loss of a *Casp3* in the *Shh* hypomorph worsened the phenotype. Note a reduction of the distal segments compared to the *Shh* hypomorph alone (inset – dorsal view shows lack of vertebrae and proximal ribs). (E) Normal embryo, ventral (top) and dorsal view (bottom). (F–H) Range of hypomorph phenotypes. Those with the least severe phenotype (F) have some disruption of the vertebrae and the most proximal ends of the proximal ribs, while the most severe phenotype is similar to *Shh* KO embryos with complete loss of vertebrae and proximal ribs (H). Phenotypes were observed in at least 3/3 animals of each genotype. *Ptch1* expression after Tamoxifen injection at E7.0 and E8.0 was assessed at E8.0 and E.9.0, respectively, see *Figure 3—figure supplement 1*. Skeletal defects could be due to alterations in somite patterning. Experiments addressing this possibility can be found in *Figure 3— figure supplement 2*.

*Figure 3 continued on next page*

*Figure 3 continued*

DOI: https://doi.org/10.7554/eLife.29144.005

The following figure supplements are available for figure 3:

**Figure supplement 1.** Expression of *Ptch1* in *Foxa2*-CRE-ERT2;*Shh*⁻/fl embryos.

DOI: https://doi.org/10.7554/eLife.29144.006

**Figure supplement 2.** Somite patterning in *Apaf1;Shh* KO embryos is similar to *Shh* KO embryos.

DOI: https://doi.org/10.7554/eLife.29144.007

the production of an alcian-blue-positive matrix (*Akiyama et al., 2002*) (*Lefebvre and Smits, 2005*). *Col2a1, Sox9* expression and alcian blue staining in control embryos extends from the vertebrae laterally approximately halfway around the chest (*Figure 4K,N,Q*). *Shh* KO embryos exhibited staining for these markers but only in a distal portion aligned under the forearms (*Figure 4L,O,R', R'*). However, in the body wall of *Shh;Apaf1* DKO embryos, no expression of *Col2a1, Sox9,* or alcian blue staining was evident (*Figure 4M,P,S,S'*). Thus, in summary, these assays demonstrate that in *Shh* KO embryos although the distal rib anlage is smaller compared to controls, differentiation proceeds normally. However in *Shh;Apaf1* DKO embryos, the rib anlagen are even smaller than seen in *Shh* KO embryos, and while some cells turn on *Sox9* and some aggregates form, they do not condense normally, and fail to differentiate.

## An agent-based simulation is sufficient to explain the phenotypes

The formation of smaller aggregates in the *Shh;Apaf1* DKO is at first confusing since the loss of cell death might be expected to result in *increased* tissue growth as has been observed in the brains of *Apaf1* and *Casp3* embryos (*Cecconi et al., 1998*; *Yoshida et al., 1998*). In contrast, while both *Apaf1* KO and *Casp3* KO animals have decreased cell death in the somites relative to normal, the thoracic skeleton does not appear dramatically overgrown (*Figure 1E*). One possibility is that the normal level of cell death in the somite is not very high and so its loss does not produce overgrowth. Another possibility is that proliferation is decreased in the absence of *Apaf1*, compensating for the increased survival and thus a sufficient number of cells to build the thoracic skeleton is maintained. To determine if this kind of compensation could produce the observed outcomes and also to better understand the even smaller cartilage anlage phenotype in *Shh;Apaf1* DKO embryos, we decided to build an agent-based simulation using NetLogo (*Wilensky, 1999*) to model rib outgrowth and patterning up to ~E12.0.

To build the simulation, we incorporated six important causal processes. These included: a Hh signal with varying concentration, variable cell death and proliferation rates, a progenitor pool that could vary in number, boundaries as would be created by surrounding tissues, and the potential effect of local cell-cell communication. We proposed that these processes predominated and therefore composed the minimal set of factors to be considered. We then represented cells as agents (called 'turtles' in NetLogo) and created an initial field of these agents randomly placed in a square to represent the location of progenitor cells within the somite. Based on our conditional ablation results (*Figure 3*), we assumed that cells are responsive to Hh signaling in an early time window. The field of cells was then programmed to change through time within a defined rectangular space according to simple parameters: for example, the levels, diffusion, and concentration of a Hh signal could be controlled. In addition, cell death rate and timing could be modulated along with cell proliferation and the number of agents in the initial field. A change in cell fate was simulated by changing the color of some undecided 'cells' to red (for proximal) or blue (for distal) at each time step with the decision determined by the amount of local Hh signal available and with an adjustable probability of conversion. To account for the reinforcement of fate by local cell-cell communication (the 'community effect' [*Gurdon, 1988*]), each cell was programmed to evaluate the local distribution of red or blue cells at each time step, and to convert to the local majority color when a local supermajority of other-colored cells surrounded it. To simulate the early stages of outgrowth, the cells were programmed to move outward as they proliferated based on their degree of crowding at each time tick, with cells only moving when their local crowding was sufficiently high and then also confined in the rectangular space. When the cells hit the far right-side end of the defined space, the

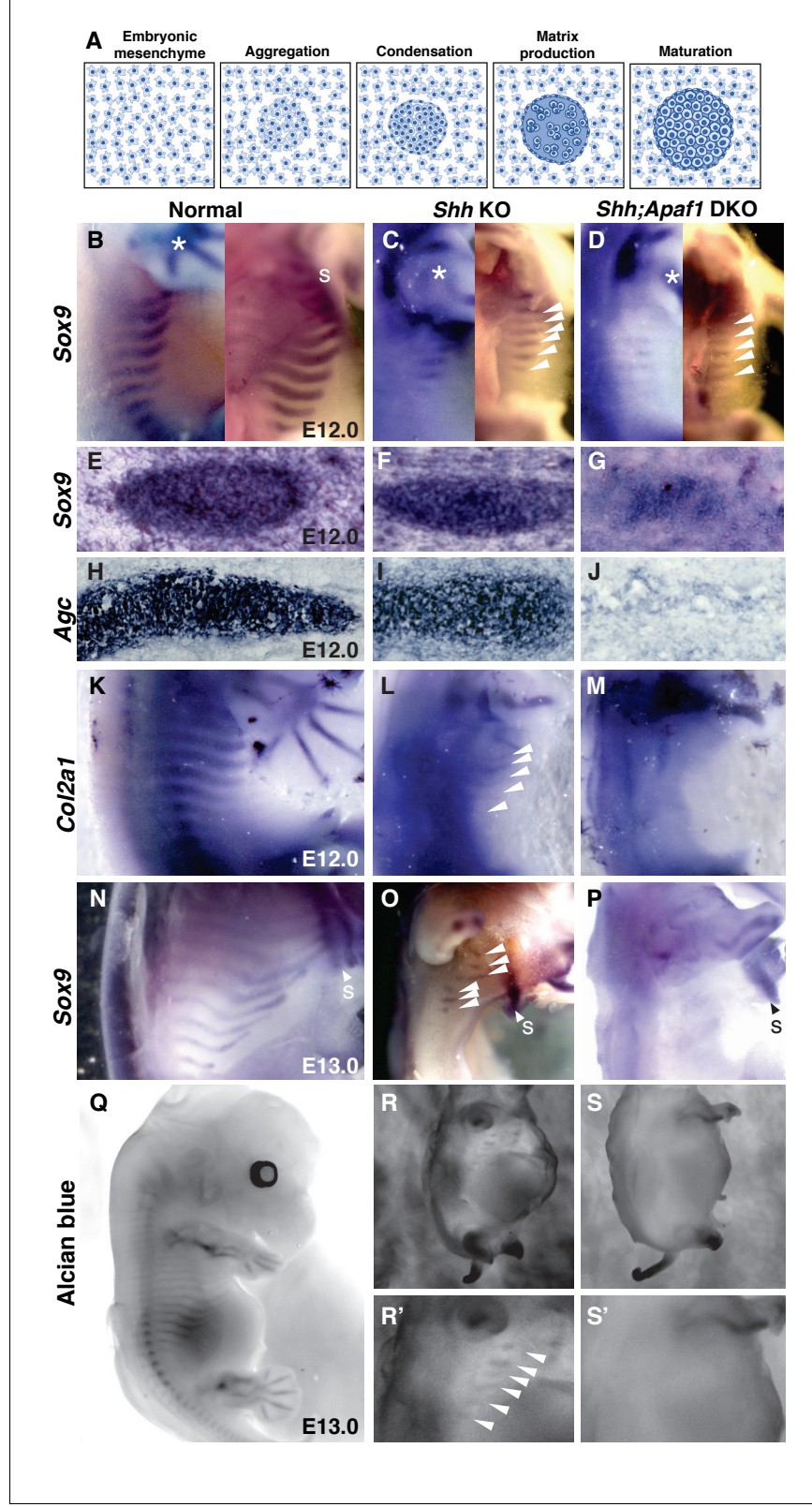

**Figure 4.** Cartilage differentiation fails in double null embryos. (**A**) Schematic showing the steps of cartilage differentiation. (**B–D**) *Sox9* expression at E12.0. First panel shows an external lateral view, second panel an internal lateral view, asterisk indicates *Sox9* expression in the limbs. (**B**) Normal embryos express *Sox9* in the ribs extending from the dorsal midline toward the sternum (**S**). (**C**) *Shh* KO embryos have reduced *Sox9* expression domain

*Figure 4 continued*

(thinner and shorter) and located distally. (D) *Shh;Apaf1* DKO embryos have a further reduced *Sox9* expression in the distal domain. (E–J) Cross-sections show *Sox9*- and *Agc*-expressing condensed cartilage forming in normal and *Shh* KO embryos while *Shh;Apaf1* DKO embryos have only a loose aggregate of *Sox9*-expressing cells and very few loosely distributed cells, if any, expressing *Agc*. (K–M) Normal embryos express *Col2a1* in a band of cells from the vertebrae toward the sternum (K). (L) *Shh* KO embryos express *Col2a1* only distally under the forearms (arrowheads). (M) No *Col2a1* expression is observed in *Shh;Apaf1* DKO embryos in the thoracic area. (N–P) Continued *Sox9* expression is observed in controls (N) at E13.0. (O) In *Shh* KO embryos, *Sox9* expression is present but only distally (arrowheads). (P) *Sox9* expression is undetectable in the entire thoracic area of *Shh;Apaf1* DKO embryos. (Q–S) Alcian blue cartilage staining of E13.0 embryos is shown in black and white for greater contrast. (Q) Cartilage anlagen in normal embryos extend from the vertebrae toward the sternum. (R, R′) *Shh* KO embryos have distally located rib cartilages (arrowheads). (S, S′) No alcian blue positive rib cartilages are observed in *Shh;Apaf1* DKO embryos. Expression patterns were consistently observed in at least 3/3 animals of each genotype.

DOI: https://doi.org/10.7554/eLife.29144.008

clock was stopped to indicate the end of outgrowth (see Materials and methods and *Supplementary file 2* for more details on the model design).

While the agent-based simulation represents a highly simplistic scenario compared to real cells that have a particular developmental history, a complex relationship with their environment, and specific migratory properties, we were impressed to find that this small set of six processes was both necessary and sufficient to simulate a wide range of phenotypes. With the model structure established, we began to set parameters to different values based on biological insight such that different phenotypes could be generated based on a final visual outcome. It became readily apparent that in order to simulate the *Shh;Apaf1* DKO, which has smaller elements than normal or even the *Shh* KO (*Figure 4D*), that even in the absence of cell death, cell proliferation and/or somite size must be reduced to obtain the predicted outcome. In order to confirm this qualitative prediction and estimate more precise parameter settings, we returned to our biological samples to measure somite size and proliferation in the different genotypes. We therefore analyzed somite-matched E9.0 embryos when multiple somites are readily visible and the forelimb bud has begun to emerge providing an internal landmark across all genotypes. Embryos were analyzed for size and for the expression of phosphorylated histone H3 (pHH3), an indicator of cells in mitosis (*Figure 5A–D*). To obtain an estimate of the relative size and proliferation rates for the different genotypes, we created a hierarchical Bayesian measurement model (see *Supplementary file 2*). In line with the observed data points (*Figure 5E*, top graph), we discovered that any difference in size between normal and *Apaf1* KO embryos was negligible and that both *Shh* KO and *Shh;Apaf1* DKO were noticeably reduced in size (*Figure 5E*, bottom graph). Furthermore, *Apaf1* KO embryos had a reduced proliferation rate which was further reduced in the absence of *Shh* (*Figure 5F*). This confirms that the loss of *Apaf1* does result in a compensatory decrease in proliferation and that the effect appears to interact with being null for *Shh* to further reduce the proliferation rate in *Shh;Apaf1* DKO embryos. Using this analysis, we then chose parameter values for proliferation and initial size for all genotype simulations equal to the median posterior sample value from the Bayesian model. Using these values (see *Table 1* and *Figure 6—figure supplements 1* and *2*), we observed that the final pattern of the simulations visually matches the expected phenotypic outcomes (*Figure 6A–C*).

## Parameters that were used to simulate the different phenotypes

Values different from normal are shaded. Values for size and proliferation rate were chosen based on the median posterior distribution from a Bayesian model, see *Supplementary file 1*.

Using an initial fixed size and our newly defined baseline parameters, we found that the simulation could represent the response to a Hh concentration gradient reliably with the percent of proximal red and distal blue cells dependent on the Hh dose and gradient steepness (*Figure 6B* and *Figure 6—figure supplement 1*). In addition, when varying the initial size, with fixed morphogen parameters, smaller initial sizes are more proximalized as a larger percentage of the cells are under the influence of high Hedgehog signaling (*Figure 6—figure supplement 2*). Interestingly, the generation of a highly distinct border between the two elements did require that cells be programmed to

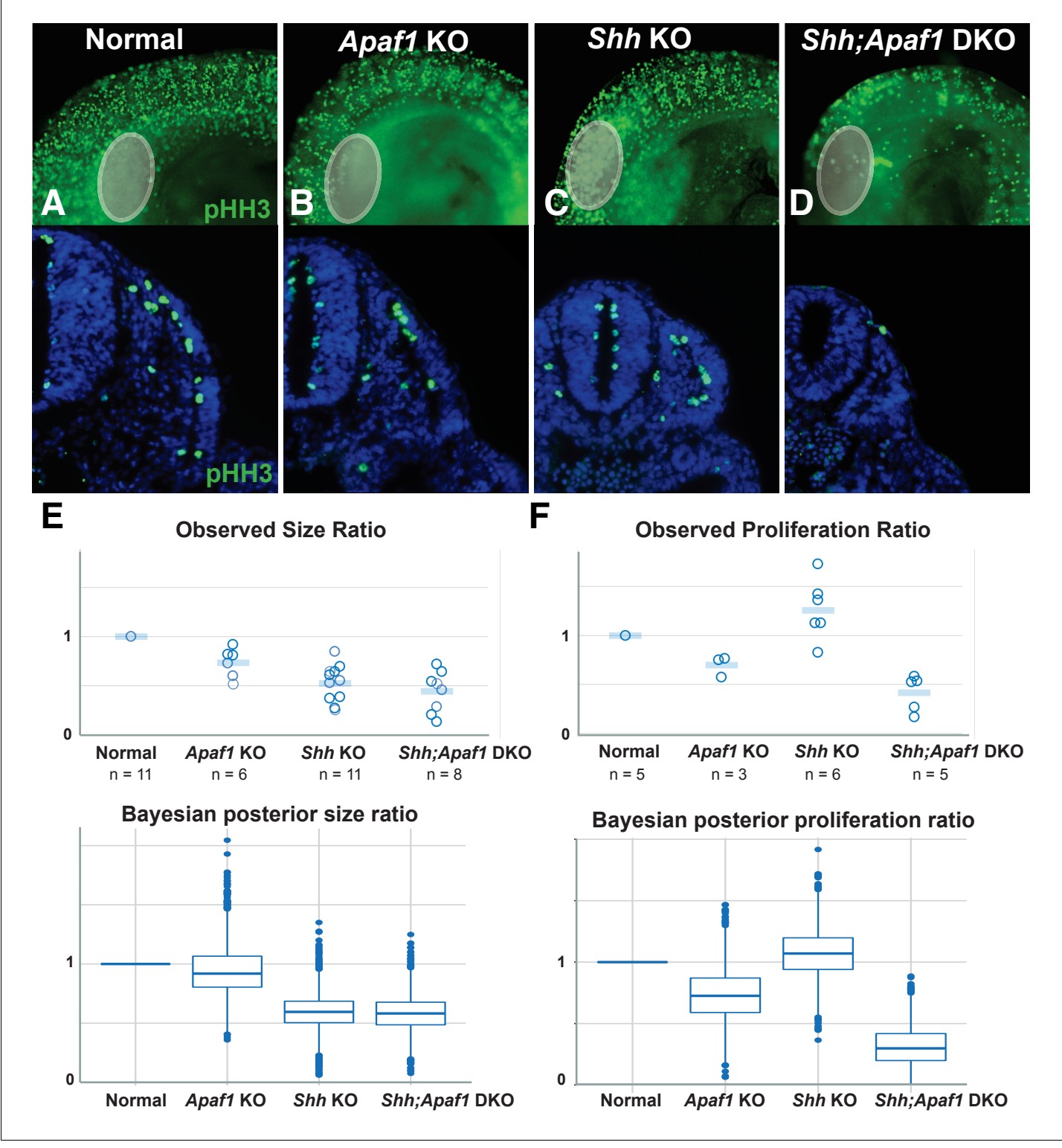

**Figure 5.** Alterations in somite size and proliferation are evident in the different genotypes. (A–D) Representative longitudinal views (top, limb bud indicated by grey oval) and transverse sections (bottom) from somite-matched E9.0 mouse embryos at equivalent locations (based on anatomical landmarks). Phosphorylated histone H3 (pHH3) immunofluorescence reveals cells in mitosis (green: pHH3; blue: DAPI). (E). (Top) Ratio of average somite size to average litter-matched controls size is plotted directly. (Bottom) The probability distribution of Bayesian estimates for size ratios is shown in box plot format. Bayesian estimates indicate that *Shh* KO and *Shh;Apaf1* DKO embryos have smaller somites than either controls or *Apaf1* KO embryos. Median estimates are *Apaf1* KO = 0.92, *Shh* KO = 0.59 and *Shh;Apaf1* DKO = 0.57, probability that the ratios are less than 1 are 0.67, 0.99, and 0.99,
*Figure 5 continued on next page*

*Figure 5 continued*

respectively. (F) (Top) Ratio of average somite proliferation rate to average litter-matched controls size is plotted directly. Normal and *Shh* KO embryos have a similar ratio of pHH3 positive to total cells in the somite (shown relative to controls). While *Apaf1* KO and *Shh;Apaf1* DKO embryos both exhibit a decrease in the ratio of pHH3 positive to total cells. (Bottom) The probability distribution of Bayesian estimates for size ratios is shown in box plot format. Bayesian estimates indicate that *Apaf1* KO and *Shh;Apaf1* DKO embryos have a reduce proliferation rate than either controls or *Shh* KO embryos. Median estimates are *Apaf1* KO = 0.71, *Shh* KO = 1.1 and *Shh;Apaf1* DKO = 0.32, probability that the ratios are less than 1 are 0.91, 0.33, and 1.0, respectively. Boxes represent 50% probability with a line at the median, plot uses standard settings for ggplot2. See also **Supplementary file 1**, **Source code 1**, and **Source code 2** for a description of the analysis of variance and the accompanying code, and **Figure 5—source data 1** for the raw data.

DOI: https://doi.org/10.7554/eLife.29144.009

The following source data is available for figure 5:

**Source data 1.** Measurements collected to create graphs in **Figure 5E and F**.

DOI: https://doi.org/10.7554/eLife.29144.010

sense the fate of local cells within their environment (the 'community effect') as predicted by early studies on muscle differentiation (**Gurdon, 1988**) (**Figure 6—figure supplement 2**). In addition, alterations in the rate of cell death or of proliferation, while profoundly impacting overall size, did not have a large effect on the ratios of cells that contributed to a proximal vs. distal element (**Figure 6—figure supplement 3**). Thus, our model displays robust reproduction of a wide range of observed phenomena (see **Video 1** for simulations for the different genotypes).

## Discussion

In contrast to the vertebrate limb, which is comprised of multiple segments and skeletal elements, the rib is comparatively simple with just two distinct proximal and distal skeletal elements. This simple organization combined with an analysis of different genetic contexts provides us with the opportunity to re-evaluate models of skeletal patterning in the context of growth. To understand how patterning arises, an important goal is to determine how cells make decisions especially considering their local environment. Agent-based modeling, by virtue of placing the burden of computation on the cell (or 'agent'), facilitates this thought process. Once a set of behaviors for agent decision-making and interaction is established, system-level patterns emerge from these interactions. This approach has been invaluable for understanding cellular phenomena such as quorum-sensing in bacterial biofilms (reviewed in [**Gorochowski, 2016**]), the arrangement of pigment cells into stripes (**Volkening and Sandstede, 2015**), and even how stem cell self-renewal could increase when developing mammary gland tissue has been exposed to radiation (**Tang et al., 2014**).

In this study, we incorporate a role for Hh concentration in determining cell fate, the influence of cell proliferation and cell death on organ size, and the interaction of cells within boundaries in order to understand how a structure emerges from local decisions. By simulating a biological system using rules that respect local decision-making, and restricting the parameters to those compatible with the

**Table 1.** Parameter values used for running the simulations

| Parameter | Variable name | Normal | *Apaf1* KO | *Shh* KO | *Apaf1;Shh* DKO |
|---|---|---|---|---|---|
| Initial size | *initsizemult* | 1 | 0.92 | 0.59 | .57 |
| Extent of Hh signal | *shh-xport* | 12 | 12 | 12 | 12 |
| Hh intensity | *shh-intensity-log* | 0.6 | 0.6 | -2 | -2 |
| Time | *nticks* | 46 | 46 | 46 | 46 |
| Probability of becoming proximal | *pRed1* | 0.4 | 0.4 | 0.4 | 0.4 |
| Probability of becoming distal | *pBlue1* | 0.4 | 0.4 | 0.4 | 0.4 |
| Peak cell death rate | *celldeathmult* | 0.3 | 0 | 1 | 0 |
| Proliferation rate | *proliferatemult* | 1 | 0.71 | 1.1 | 0.32 |
| Duration of cell death | *cdduration* | 30 | 30 | 30 | 30 |

DOI: https://doi.org/10.7554/eLife.29144.011

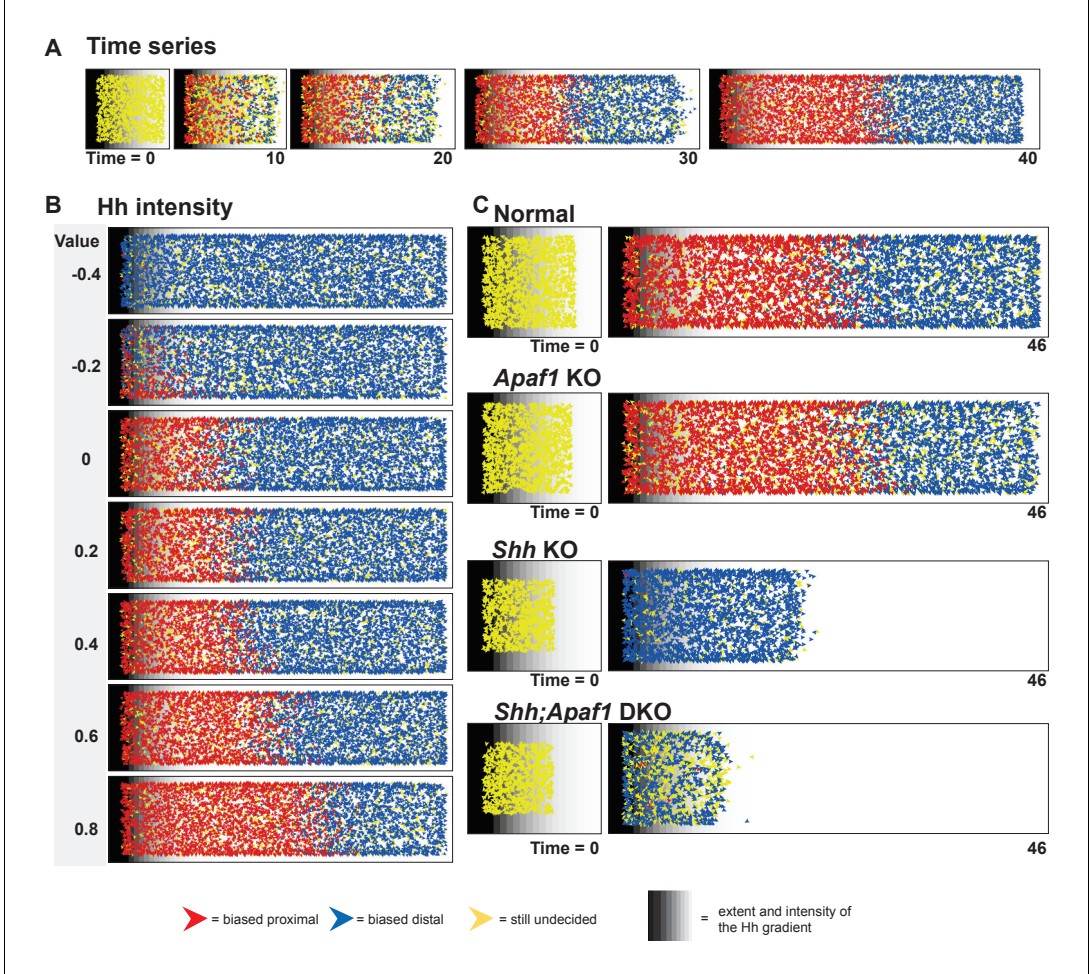

**Figure 6.** An agent-based model simulates rib proximal-distal patterning and outgrowth. (A) A representative time series for the normal condition. The model simulates rib development up to E12.0 just prior to differentiation of the condensations. The red and blue agents represent cells biased to become part of a proximal or distal element, respectively. Yellow indicates that a fate decision has not yet been made. Gray stripes indicate the extent and intensity of the Hh gradient. (B) Outcome of adjusting Hh intensity without adjusting the gradient pitch using baseline parameter settings for the normal genotype. (C) Representative starting points and outcomes that mimic the phenotypes. Normal was simulated with a moderate Hh gradient intensity, mild cell death, and a moderate proliferation rate. The loss of *Apaf1* was simulated by decreasing cell death to zero, decreasing the proliferation rate, and moderately decreasing the number of initial progenitors; loss of *Shh* was simulated by decreasing the Hh intensity, decreased progenitor pool size, slightly increased proliferation rate, and increased cell death. The absence of both *Shh* and *Apaf1* was simulated with no cell death, an even smaller initial pool was used and an overall decrease in cell proliferation. More undecided (yellow) agents can be found in this case because proliferation is very low and the model assumes decisions are made at cell division. Over time, these cells eventually become blue due to low Hh levels. A detailed explanation of the model can be found in *Supplementary file 2*. The parameter values used to generate the outcomes are listed in *Table 1*, and the outcome of additional simulation runs while varying specific parameters can be found in *Figure 6—figure supplements 1*, *2* and *3*. Example video simulations can be found in *Video 1* and the code to run the simulations can be found in *Source code 3*.

DOI: https://doi.org/10.7554/eLife.29144.012

The following figure supplements are available for figure 6:

**Figure supplement 1.** Outcome from multiple simulations for each genotype.
DOI: https://doi.org/10.7554/eLife.29144.013

**Figure supplement 2.** The impact of altering the nearest neighbor effect and the initial size on simulation outcome
DOI: https://doi.org/10.7554/eLife.29144.014

**Figure supplement 3.** Outcome after varying the proliferation and cell death parameter values
DOI: https://doi.org/10.7554/eLife.29144.015

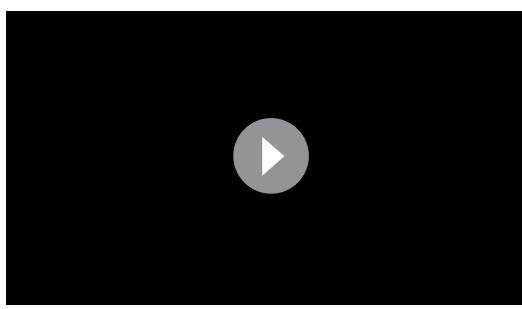

**Video 1.** Example simulations using the parameters listed in *Table 1*
DOI: https://doi.org/10.7554/eLife.29144.016

observed phenotypes in our genetically modified embryos, we infer a minimum set of rules that can be used to explain how pattern arises. We show that patterning can happen early when cell populations are small and thus the distances required to interpret a Hh gradient are not too great. In addition, we infer how size is regulated via mechanisms that control cell number such as initial progenitor number and proliferation rate. Through this analysis we have generated an overall picture for how patterning occurs that can be used to explain the different phenotypes observed (*Figure 7*).

## The role of Hh signaling is multi-functional and early

As observed in many other contexts (*Briscoe and Small, 2015*), the role of Hh signaling is multi-functional and likely required for many processes during rib development including: (1) sclerotome induction, (2) maintenance of cell survival, and (3) proximal-distal specification.

1. Hh signaling is required for sclerotome induction as Shh is necessary and sufficient to specify the sclerotome and also necessary to maintain it (*Fan and Tessier-Lavigne, 1994*; *Borycki et al., 1998*; *Marcelle et al., 1999*). Furthermore, in the absence of the receptor *Smo*, sclerotome fails to form as indicated by lack of *Pax1* expression (*Zhang et al., 2001*). Based on the *Smo* null phenotype, *Shh* KO embryos might be expected not to form ribs at all. However, although the costal cartilages are reduced in size, they still form and sclerotome markers are still expressed. The presence of these structures may be due to the ectopic expression of *Ihh* observed in the foregut of *Shh* KO embryos (*Zhang et al., 2001*; *Fogel et al., 2008*). This Hh source could also explain the presence of some *Pax1* expression in *Shh* null embryos and its later loss as cells move away from the midline.

2. Hh signaling is important for cell survival since cell death in the somite is vastly increased in *Shh* KO embryos. This role is likely transient and starts early (prior to E9.0) as indices of cell death (LysoTracker and TUNEL) are high at E9.0 and by E11.5 have decreased with staining only apparent in the caudal yet undifferentiated somites. Together, both the activities of sclerotome induction and cell death, because they happen early on, determine the size of the initial cell progenitor pool. As long as the proliferation rate is normal or at least sufficient, a modest decrease in initial progenitor pool may have little impact on patterning (as seen in the *Apaf1* KO).

3. While we cannot unequivocally rule out other models for proximal-distal patterning, a role for high and low levels of Hh signaling specifying proximal and distal fates, respectively, is supported by the skeletal pattern observed when Hh signaling is mildly compromised. In particular, the range of phenotypes observed in the *Foxa2*-CRE;*Shh* hypomorphs suggests that mild decreases in Hh signal results in partial reduction in the number of cells specified as proximal, while severe decreases in Hh signal results in massive reduction in the number of cells specified as proximal. In addition to the phenotypic evidence, our gene expression data is also supportive of this dose dependence as there appear to be sclerotome populations that are more and less sensitive to a Hh signal. For instance, considering the presence of *Pax1* in *Shh* KO embryos and its later loss, we suggest that the *Pax1*-expressing population makes a significant contribution to the proximal rib segment. *Foxc2* is also expressed in the absence of *Shh*, however, *Foxc2* expression is not lost later. We suggest that this remaining *Foxc2*-expressing population can be induced/maintained by low Hh signaling and represents cells that contribute to the distal segment that still forms in *Shh* KO embryos. How high vs. low Hh signaling specifies proximal vs. distal fates, respectively, is not clear but may involve different combinations of Gli transcription factors as has been elegantly demonstrated in neural tube patterning (*Peterson et al., 2012*; *Cohen et al., 2015*). Loss of *Gli2* and *Gli3* can influence the ability of the somite to respond to Hh signaling and turn on *Pax1* (*Buttitta et al., 2003*); however, more studies will be needed to understand the details of this at the transcriptional level.

It remains an open question as to when proximal-distal specification occurs during early skeletal development. Two competing hypotheses, based on studies in limb development, include the Early

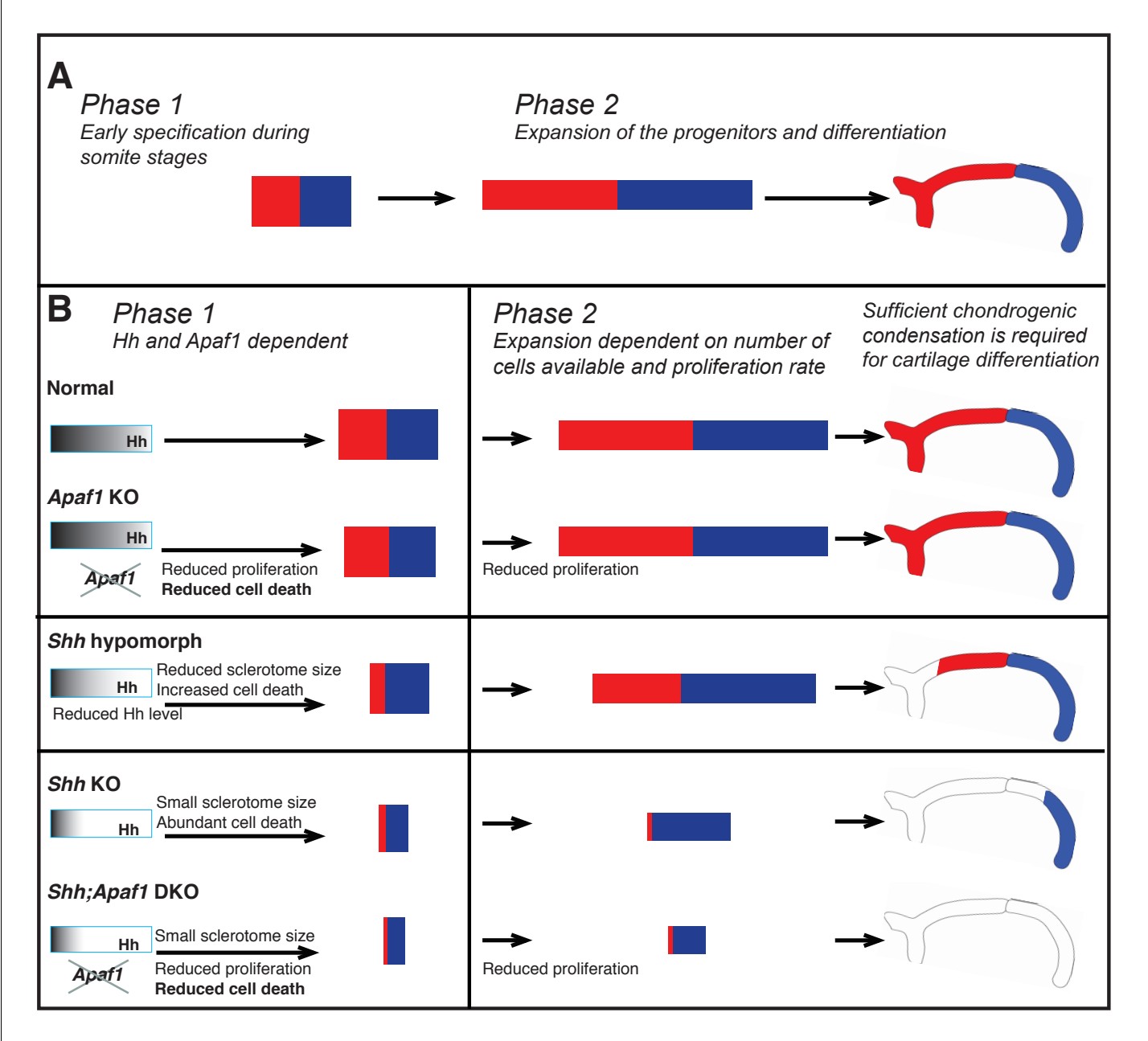

**Figure 7.** Summary of rib proximal-distal patterning: early specification and expansion. (**A**) Based on insight from our computational simulations, we propose a qualitative description of normal development in terms of two effective phases: early specification (Phase 1) and expansion (Phase 2). Specification appears to occur early (during somitogenesis), while the population of cells resides within the range of the Hh gradient, and then expansion and differentiation occur during embryo growth with proximal and distal fates being maintained by some sort of community effect independent of the influence of Hh. (**B**) We suggest that during normal development, Phase 1 involves the biological processes of sclerotome induction, cell death, and proximal-distal patterning, while Phase 2 involves expansion of cell number dependent on: (1) the events of Phase 1, (2) the proliferation rate, and (3) the constraints of the space in which expansion happens (body size). In the absence of *Apaf1*, development is largely normal as a slighly smaller initial progenitor pool due to decreased cell proliferation is mostly compensated for by decreased cell death. In *Shh* KO embryos, Hh signaling is greatly reduced during Phase 1 and thus the number of cells induced to become sclerotome is reduced (much smaller initial size). In addition, low Hh signaling specifies predominantly distal sclerotome. Despite a high level of cell death, a sufficient number of cells proliferate and are able to differentiate into the most distal segment during Phase 2. When Hh signaling is not as dramatically reduced (as in *Shh* hypomorphs) distal and some proximal specification can occur. In *Shh:Apaf1* DKO embryos, however, along with reduced sclerotome due to both loss of *Shh* (smaller initial size), cell proliferation in the somite is also further reduced. Thus, ultimately fewer skeletal progenitors are generated during Phase 2. These embryos

*Figure 7 continued on next page*

*Figure 7 continued*

are still able to form small distal aggregates; however, they are smaller than those found in *Shh* KO embryos, fail to condense or differentiate, and no rib skeletal elements form.

DOI: https://doi.org/10.7554/eLife.29144.017

Specification Model and the Progress Zone Model (reviewed in [*Mariani and Martin, 2003*; *Bénazet and Zeller, 2009*]). In the context of the rib, we favor an early specification scenario where proximal-distal specification occurs when the cells have not yet migrated long distances away from the midline and the progenitor pools are quite small. Therefore, the transport of the Hh protein can occur over a short period of time and transport over long distances is not required to explain the results (Phase 1). Then as the embryo grows, the specified compartments would expand laterally (Phase 2) (*Figure 7*). In recent years, this idea of early proximal-distal specification followed by expansion to establish the initial broad proximal and distal fields has been gaining traction in the field of limb development ([*Roselló-Díez et al., 2011*; *Cunningham and Duester, 2015*] and reviewed in *Mariani [2010]*) and seems easily applicable in the case of rib patterning. Support for a role for Shh signaling at very early somite stages (prior to E9.0), comes from our experiments where tamoxifen administration at E7.0 resulted in thoracic skeletal defects similar to *Shh* KO embryos (*Figure 3*), while administration after E8.0 or later was not sufficient to generate a thoracic phenotype. These results also align with previous studies suggesting that Hh signaling is only initially required at presomitic mesoderm stages and that subsequently BMPs are important in axial skeletal growth to maintain a chondrogenic regulatory loop (*Zeng et al., 2002*; *Stafford et al., 2011*).

In summary, Hh signaling plays several major roles—influencing the number of cells induced to become sclerotome, reducing the number of cells undergoing cell death, and causing rib progenitors to adopt a proximal vs. distal fate. The effect of Hh is likely early, during somite stages, and relatively short-range. Indeed, within our agent-based simulations, the length scale over which Hh concentration can influence cell fate is approximately the same as the size of the progenitor pool and most cells 'decide' their fate relatively early. We suggest that cells making *late* decisions become distal because they experience a very low Hh signal or become proximal due to influence from surrounding proximal cells.

## Apaf1 influences cell death and proliferation during rib development

Considering the prevalence of programmed cell death during development (*Hardy et al., 1989*), the relatively minor developmental defects seen when null for apoptosis genes has been surprising. One explanation has been that apoptosis genes can compensate for each other and/or that multiple genes need to be removed in order to see profound phenotypes (*Nagasaka et al., 2010*). Another possibility is that cell death still occurs but by other cell death pathways (*Yuan and Kroemer, 2010*). However, another explanation is that the loss of these genes *does* block normal cell death and that the proper cell number is re-established by compensatory mechanisms that decrease cell numbers such as decreased proliferation (*Cecconi et al., 2004*; *Huh et al., 2004*). In the context of rib development, in the absence of *Apaf1*, somite cell death is inhibited. When modeling this in our agent-based simulation, the proliferation rate must be moderately reduced to compensate for reduced cell death in order to achieve normal skeletal size. This prompted us to look more carefully at the proliferation rate and we did in fact see that it is decreased when *Apaf1* is lost (*Figure 5*). We suggest that further perturbations (loss of *Shh*) influences the initial sclerotome cell number and this combined with a further reduction in proliferation rate leads to the more severe defects seen in the *Shh*; *Apaf1* DKO embryos.

An interesting question for future research is determining the mechanism by which loss of *Apaf1* results in reduced proliferation. Two potential mechanisms are (1) the direct abrogation of the cell cycle machinery when the apoptosis pathway is blocked or (2) the indirect effect of a reduced number of dying cells that can release growth factors. Compensatory proliferation with dying cells releasing growth factors into their environment has been observed in other contexts (*Fan and Bergmann, 2008b*; *Jäger and Fearnhead, 2012*). Studies from *Drosophila* reveal that the signaling pathways involved in compensatory proliferation differ depending on the tissue and the developmental state of the tissue. Highly proliferating tissues have been shown to induce Tgfβ and Wnt homologues

(*Pérez-Garijo et al., 2004*; *Ryoo et al., 2004*), while differentiating tissues activate Hh signaling (*Fan and Bergmann, 2008a*). In the sclerotome, dying cells could be releasing growth factors that are sufficient to maintain the proliferation of the somite. In our studies, an increase in cell death in the *Shh* KO does correlate with a possible small increase in proliferation above normal (*Figure 5*). Then, in the absence of cell death (*Apaf1* KO), perhaps these growth factors are not released and this is a mechanism by which proliferation rate is decreased from normal.

## Skeletal development as biphasic

Based on our combined genetic analysis and agent-based simulations, it can be useful to think of rib development in terms of two main phases (*Figure 7*). During Phase 1, which starts as the presomitic mesoderm is forming into somites (E8.0-E10.0), the dominating activities are the induction of sclerotome and the establishment of proximal/distal cell identity mediated by Hh signaling. While during Phase 2 (E10.0 and onwards), the dominating activity is expansion as cell identity is maintained. The transition between phases likely occurs gradually as the skeletal elements increase in size. The ultimate size of a skeletal element at birth then, is determined by the number of cells left after Phase 1, along with an expansion of that population during Phase 2. Based on our agent-based simulations, the proliferation rate not only influences cell numbers early but, during Phase 2 can also have a profound impact on segment size because cell number increases exponentially. In the *Shh;Apaf1* DKO embryos, the decrease in transient cell death likely has a minor contribution compared to the decrease in sclerotome induction (due to loss of Hh signaling) and the decreased proliferation rate throughout somite development (due to loss of *Apaf1*).

## Final pattern is also influenced by the ability of progenitors to differentiate

Even if cartilage specification occurs, successful differentiation is still needed to achieve the final skeletal pattern. In *Shh* KO embryos, even though there are few cells participating, they are still able to aggregate, condense, and differentiate into matrix-producing cartilage tissue. However, in *Shh; Apaf1* DKO embryos, while sclerotome is present and some cells begin to express *Sox9*, the population pool is even smaller, condensation fails to occur and differentiation does not proceed (*Figure 4*). While cooperation of both Hh signaling and Apaf1 could be required for chondrocyte differentiation, another possibility is that chondrogenesis fails due to an insufficient number and density of cells. In vitro studies have shown that robust proliferation along with high cell number and density are very important for the production of cartilage matrix and chondrocyte differentiation (*Denker et al., 1999*; *DeLise et al., 2000*; *Malko et al., 2013*). Thus, differentiation in vivo may also be highly dependent on these characteristics. In *Shh;Apaf1* DKO embryos, a smaller initial size, along with decreased proliferation could result in a density of *Sox9*-expressing cells that is insufficient for differentiation to occur and thus condensation fails resulting in a more severe phenotype. The requirement for a sufficient number of cells may also account for the variation in distal rib phenotypes seen in *Shh* KO embryos (*Figure 1*).

A high cell density may result in the production of growth factors at sufficient levels for differentiation. In particular, BMP and TGFβ pathways play prominent roles in differentiation (*Wang et al., 2014*) with the cartilage cells *themselves* producing the critical ligands during pre- and early condensation stages. BMP signaling is required for chondroprogenitor aggregation and condensation in vitro (*Barna and Niswander, 2007*) and in vivo via smad1/5/8 signaling (*Pizette and Niswander, 2000*; *Retting et al., 2009*). The application of TGFβ protein in vitro to promote differentiation has been long appreciated (*DeLise et al., 2000*), while TGFβsignaling via smad2/3 has been shown to be required for the progression of chondroprogenitors toward differentiation in vivo (*Wang et al., 2014*). Interestingly, smad4, the common smad protein involved in *both* the BMP and TGFβ response, has been shown to be required specifically within pre-cartilage cells. Without *Smad4*, *Sox9*-expressing cartilage cells fail to condense, fail to undergo differentiation, and any condensations that do form are not maintained. In addition, in the absence of the ability to respond to differentiation signals, they appear to adopt a connective tissue fate (*Bénazet et al., 2012*). Thus, BMPs and TGFβ along with other pathways (reviewed in (*DeLise et al., 2000*) are likely players in the density-dependent differentiation of chondroprogenitors. Interestingly, this autocrine mechanism for cartilage differentiation has been modeled by Newman and colleagues (*Christley et al., 2007*).

Using a discrete grid-based 2D multiscale computational model, they simulate cartilage condensation and differentiation considering cell number, cell behavior, and importantly TGFβ-mediated cell-cell signaling. As our model ends prior to differentiation, this model could follow nicely as a sequel.

Taken together, we suggest that during rib skeletal development, cells respond to local cues in a way that can be usefully modeled through a series of simple rules. During normal development, we propose that a compensatory feedback loop between the apoptotic pathway and cell proliferation plays a critical role in achieving the proper number of skeletal progenitor cells for cartilage differentiation. The precise molecular mechanism that underlies this balance may involve growth factor signaling but this is still to be discovered. Likewise, it will be important to determine if compensatory proliferation occurs in other embryonic contexts when programmed cell death is blocked. One of the most interesting issues for future investigation is to discover the molecular mechanism by which Hh concentration specifies proximal and distal identity and importantly the timing during which this happens. In addition, the precise mechanism by which refinement of pattern occurs is unknown and future studies could determine whether local signaling from the surrounding majority cells (community affect) plays a role or whether some other mechanism such as reciprocal inhibition or cell rearrangement and assortment is involved.

Beyond the specifics of rib development, the use of mathematical and computational models can be extremely useful for making testable predictions. In particular, an agent-based approach allows the modeling of developing tissues while taking into consideration the behavior and fate decisions of individual cells based on local information. Furthermore, agent-based modeling is compatible with other classical differential equation-based modeling, such as reaction-diffusion and finite-element modeling which have been previously used to model limb development and other tissues (*Zhang et al., 2013*; *Lau et al., 2015*). In the future, these combined methods could be used to generate more precise multi-scale models (*Zhu et al., 2010*) which could be used not only to better understand the emergence of pattern in laboratory organisms but also to understand how changes in cell behavior during evolution could generate new and diverse forms among different species.

## Materials and methods

### Generation of *Shh;Apaf1* and *Caspase3;Shh* DKO mice

To generate *Apaf1* and *Shh* double null embryos, females heterozygous for *Apaf1* (RRID:MGI: 3783548) and homozygous for a *Shh* 'floxed' conditional allele (RRID:IMSR_JAX:004293) (*Apaf1*$^{-/+}$; *Shh*$^{fl/fl}$) (*Yoshida et al., 1998*; *Lewis et al., 2001*) were crossed with males ubiquitously expressing the CRE enzyme (RRID:IMSR_JAX:003376) and heterozygous for both *Apaf1* and *Shh* (*Actb*-CRE; *Apaf1*$^{-/+}$; *Shh*$^{-/+}$) leading to the production of *Shh:Apaf1* DKO embryos at a 1 in 8 frequency (12.5%). Heterozygous embryos were used as controls. A standard cross (*Shh*$^{+/-}$; *Casp3*$^{+/-}$ X *Shh*$^{+/-}$; *Casp3*$^{+/-}$) was established to create *Shh;Casp3* DKO embryos at a 1 in 16 frequency (6.25%) (RRID:IMSR_EM: 06370) (*Woo et al., 1998*). Conditional *Shh;Apaf1* DKO or *Shh;Casp3* DKO mice were produced using a Tamoxifen-inducible *Foxa2*-CRE-ERT2 line (RRID:IMSR_JAX:008464)(*Park et al., 2008*). Tamoxifen (Sigma (St. Louis, MO); 3 mg/mouse) and progesterone (Sigma; 1.5 mg/mouse) were administered by intraperitoneal injection 7.0–10.0 days of gestation. Embryos were genotyped by PCR. All animal procedures were carried out in accordance with approved Animal Care and Use Protocols at the University of Southern California.

### Histology and gene expression assays
#### Bone and cartilage staining
Embryos were collected E12.0–18.5 and fixed in 95% EtOH. Skeletal preparations were carried out following a standard protocol (*Rigueur and Lyons, 2014*).

#### Sectioning
Tissue was cryo-protected and embedded in Tissue-Tek (Torrance, CA) OCT compound in a peel-a-way embedding mold for frozen sectioning.

## RNA in situ hybridization

Embryos were harvested and immersion-fixed in 4% paraformaldehyde. Digoxigenin-conjugated antisense riboprobes were prepared (Roche (Switzerland), cat# 11175025910) and whole-mount or section RNA in situ hybridizations were color developed using NBT/BCIP (Roche, cat # 11681451001) according to previously established protocols.

## Agent-based modeling

An Agent-Based Model (ABM) was developed using the NetLogo system version 6.0 (*Wilensky, 1999*). The model operates by initially creating a field of agents representing cells (called 'turtles' in NetLogo) and then evolving this field through time according to simple rules. A varying concentration of secreted Hh is modeled by using a Gaussian type curve with peak at $x = -10$ and a variable width and height. The agents are initially randomly placed in a square block between $x = 0$ and $x = sqrt(N/1200)*14$ and $y = \pm(sqrt(N/1200)*14)/2$. They divide with base probability of 0.05 per time step and die with base probability 0.05 per time step, and the death rate is multiplied by a variable-relative rate and a time-varying curve using a Gaussian with a variable-duration width in time. At each cell division, there is a probability to convert from yellow (unspecified) to red (proximal) or blue (distal) and this is based on the local concentration of Hh. Once converted (biased in fate), 'cells' ignore the Hh concentration (mimicking the window of competence to respond), but at each time point, the local concentration of red or blue cells is assessed and the cell is programmed to convert to the local majority color when a local super-majority of other-colored cells surround it.

As the cells proliferate, the cells spread away from regions of high local density maintaining a density of 4–6 cells per patch. The net effect is that the cell field expands as the cells divide. The cells are prevented from moving through the boundary of the spatial field. When the cell field hits the farthest column of patches the clock is stopped, otherwise the simulation continues for the specified number of time ticks to models a slowing in growth rate as structures reach the size limit of the animal. Without this boundary effect the number of cells would grow exponentially in time and without bound.

By varying the initial cell field size, the Hh concentration in space, the time duration, the coefficients that determine yellow-red and yellow-blue conversion probabilities, the cell death intensity, the proliferation rate, and the duration of the cell death in time, we can model a variety of conditions. See *Supplementary file 2* for further details on the model. Code to run the simulations can be found in *Source code 3* and run after downloading the free NetLogo program at: https://ccl.northwestern.edu/netlogo/.

## Somite cell death, size, and proliferation rate

### Cell death

Lysosomal activity (which correlates with cell death) was assayed with the dye LysoTracker-RED (Thermo Fisher Scientific, Waltham Ma) (*Fogel et al., 2012*). The LysoTracker staining patterns were confirmed with TUNEL analysis (In Situ Cell Death Detection Kit, Roche).

### Proliferation and somite size

Anti-phospho-Histone H3 antibodies (06-57,0 Millipore Sigma, Burlington, Ma) were used in 1:500 concentration, followed by an Alexa Fluor 488 goat anti-rabbit IgG (H + L) antibody (A11008, Thermo Fisher Scientific) at a 1:250 concentration. To measure somite size, an ROI was defined from images of the center of the somite and the pixels were counted in Adobe Photoshop or Image J. To determine proliferation rates, the pHH3-positive cells were counted vs. DAPI-positive nuclei within the somite. To display the data from each embryo, the average values compared to the control in a set was calculated using Excel and graphed in *Figure 5E,F* (top). While this method allows the display of raw data (as ratios of averages) it does not lend itself to statistical analysis. To generate a best estimate of the proliferation rate and initial size of somites in each genotype relative to normal, a hierarchical Bayesian model was developed and fitted to the collected data using Stan via the rstan package in R (*Carpenter et al., 2017*). Median posterior values for the relative size and relative proliferation rate were used in the prototype simulations. For more details see *Supplementary file 1*. The R script and rstan code can be found in *Source code 2* and *Source code 3*. For most samples,

between 6 and 8 somites were measured; however, for a few samples, only 2–4 somites were available. A table of all data collected can be found in *Figure 5—source data 1*.

## Acknowledgements

We gratefully thank Gail R Martin and Andrew P McMahon for advice during the initial stages of this project, Cheng-Ming Chuong, Leonardo Morsut, Michael Elowitz, and Steven Vokes for comments on manuscript drafts, and Audrey Izuhara, Ashlie Muñoz, and Christy Furukawa for technical assistance. This work was supported by the University of Southern California (to FVM and IKM), NIH NIAMS (to FVM) and by a CIRM training grant (to JLF).

## Additional information

### Funding

| Funder | Grant reference number | Author |
| --- | --- | --- |
| California Institute for Regenerative Medicine | Postdoctoral Fellowship | Jennifer L Fogel |
| University of Southern California | | Jennifer L Fogel<br>Daniel L Lakeland<br>In Kyoung Mah<br>Francesca V Mariani |
| National Institutes of Health | AR064462 | Francesca V Mariani |
| National Institutes of Health | AR069700 | Francesca V Mariani |

The funders had no role in study design, data collection and interpretation, or the decision to submit the work for publication.

### Author contributions

Jennifer L Fogel, Conceptualization, Data curation, Formal analysis, Investigation, Methodology, Writing—original draft, Writing—review and editing; Daniel L Lakeland, Conceptualization, Software, Formal analysis, Investigation, Visualization, Methodology, Writing—review and editing; In Kyoung Mah, Validation, Investigation; Francesca V Mariani, Conceptualization, Resources, Data curation, Software, Formal analysis, Supervision, Funding acquisition, Validation, Investigation, Visualization, Methodology, Writing—original draft, Project administration, Writing—review and editing

### Author ORCIDs

Francesca V Mariani http://orcid.org/0000-0003-1619-8763

### Ethics

Animal experimentation: This study was performed in strict accordance with the recommendations in the Guide for the Care and Use of Laboratory Animals of the National Institutes of Health. All of the animals were handled according to approved institutional animal care and use committee (IACUC) protocols (#11152) of the University of Southern California.

### Decision letter and Author response

Decision letter https://doi.org/10.7554/eLife.29144.024
Author response https://doi.org/10.7554/eLife.29144.025

## Additional files

### Supplementary files

• Source code 1. R code to process date in *Figure 5—source data 1*
DOI: https://doi.org/10.7554/eLife.29144.018
• Source code 2. Rstan Code to calculate the estimation

DOI: https://doi.org/10.7554/eLife.29144.019

- Source code 3. NetLogo code to run the simulation

DOI: https://doi.org/10.7554/eLife.29144.020

- Supplementary file 1. Description of the Bayesian method to estimate the median size and proliferation outcomes for each genotype

DOI: https://doi.org/10.7554/eLife.29144.021

- Supplementary file 2. Detailed description of the NetLogo model design

DOI: https://doi.org/10.7554/eLife.29144.022

- Transparent reporting form

DOI: https://doi.org/10.7554/eLife.29144.023

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
