## [Decision Letter]

[Editors’ note: the authors were asked to provide a plan for revisions before the editors issued a final decision. What follows is the editors’ letter requesting such plan.]

Thank you for submitting your article "A minimally sufficient model for rib proximal-distal patterning based on genetic analysis and agent-based simulations" for consideration by *eLife*. Your article has been reviewed by three peer reviewers, and the evaluation has been overseen by a Reviewing Editor (Lee Niswander) and Marianne Bronner as the Senior Editor. The following individuals involved in review of your submission have agreed to reveal their identity: Linus Schumacher (Reviewer #3).

The reviewers have discussed the reviews with one another and the Reviewing Editor has drafted this decision to help you prepare a revised submission. However, as you will see, the work necessary to address the concerns of the reviewers may take more than the two months we normally allow for return of a revised manuscript. If you feel you can address these issues in a reasonable length of time, please draft a response letter to the editor and reviewers in which you outline the work you are prepared to undertake and a time table for its completion. The editor and reviewers will consider your response and follow with recommendations.

Summary:

This manuscript from the Mariani lab focuses on an understudied aspect of development, rib formation, using mouse genetic mutants and simulation modeling to address how proximal-distal patterning of the ribs is specified during embryogenesis. The phenotype being studied and the agent-based model to generate hypotheses to be tested in vivo are interesting and informative. A model of early Hh-mediated specification followed by expansion is invoked. Although this specification-expansion mechanism has been shown in the limb and neural tube, the current results lay an excellent framework in which to conceptualize, and model with a few simple rules, various processes from graded morphogen signals to cell death and proliferation rate in conjunction with bi-phasic specification and expansion in the formation of complex structures

Despite the enthusiasm of the reviewers for the biological question and the modeling, a number of concerns were raised and it is felt that the results and modeling only address a subset of the complex phenotype and hence the manuscript remains incomplete and only a partial explanation.

Essential revisions:

The major difference between the Shh and the Shh/Apaf1 KO mouse is the absence of distal rib cartilages after e12.5. The agent-based model ends at eE12.5, so does not take this into account. While the modeling may show decreased "distal cell" number in the Shh/Apaf1 mutant condition (it is hard to actually appreciate this outcome), the observable *Sox9*-expressing condensations in the distal region suggest the initial stages of chondrogenesis are "ok." Combined histological, molecular (ISH), cellular (cell death by an apoptosis-independent mechanism?) analyses may help elucidate the problems within the distal cartilage anlagen. It seems that the authors favor an idea that the pool of chondroprogenitors is so reduced that differentiation does not take place. The authors speculate the proliferation drops due to over–compensation but this is not a very satisfying hypothesis.

An unexplained aspect is that the Apaf1 somite is reduced in size by 25% around the time that Shh-mediated specification by Shh from the notochord and floor plate should be occurring. This would seem to predict that Shh could act on a greater portion of the somite and hence could specify a greater number of cells to a rib fate or lead to an increase in proximal rib fate as in Figure 6, yet this is not observed.

Another aspect that is not explained by the model is the spatial patterning of the two segments in a Shh KO or Shh;Apaf1 dKO wherein distal chondroprogenitor are seen distally but not proximally and in the Shh KO the distal cartilage is formed next to the sternum in the absence of proximal rib cells (or proximal chondroprogenitors).

If a preprint cannot be cited, the authors should provide arguments as to why the parameters and the structure of the model are biologically plausible. Methodological details of the model are insufficiently stated, and while the code is provided, this hinders reproducibility and assessment of model justification.

Furthermore, it is not always clear in which way model outcomes and experiments were compared. For example, what constitutes "replicating the Apaf1 KO" given that this is a very small or no effect? If the point is simply that no change is observed when cell death and proliferation are lowered in concert, that is to be expected. It's fine to illustrate that with a simulation, but the point needs to made more clearly.

Was the choice of parameter perturbations to replicate gene KO an informed one, or was the parameter space systematically searched to find "best matches?

The manuscript could be strengthened by emphasizing the interplay between simulations and experiments more strongly and sooner. As it stands, there is a lot of "classic" experimental work to get through before getting to the simulations, and even then, the prediction of the simulations (about somite size and proliferation rate in the DKO) is not very clear in the figure.

Quantification of the *Sox9* domain at E12.5 in whole mount was performed, but the data are not shown and the method is not described.

What is the positive control for Apaf1 KO working, given the effect is hardly seen here?

The method for quantifying the cross–sectional area is not described. How were the sections selected for analysis, what is the sample size for each genotype?

pHH3 does not appear to label many cells in Figure 7, so the biological meaning of the changes are not clear. How many sections/embryo were analyzed? How many embryos were analyzed for each genotype?

The manuscript could be shortened overall (especially the Introduction and Discussion section).

[Editors’ note: formal revisions were requested, following approval of the authors’ plan of action.]

Thank you for choosing to send your work entitled "A minimally sufficient model for rib proximal-distal patterning based on genetic analysis and agent-based simulations" for consideration at *eLife*. Your plan has been considered by a Senior Editor and a Reviewing editor, and we are prepared to consider a revised submission with no guarantees of acceptance.

---

## [Author Response]

[Editors’ note: what follows is the authors’ plan to address the revisions.]

Essential revisions:The major difference between the Shh and the Shh/Apaf1 KO mouse is the absence of distal rib cartilages after e12.5. The agent-based model ends at eE12.5, so does not take this into account. While the modeling may show decreased "distal cell" number in the Shh/Apaf1 mutant condition (it is hard to actually appreciate this outcome), the observable Sox9-expressing condensations in the distal region suggest the initial stages of chondrogenesis are "ok." Combined histological, molecular (ISH), cellular (cell death by an apoptosis-independent mechanism?) analyses may help elucidate the problems within the distal cartilage anlagen. It seems that the authors favor an idea that the pool of chondroprogenitors is so reduced that differentiation does not take place. The authors speculate the proliferation drops due to over–compensation but this is not a very satisfying hypothesis.

The fact that chondroprogenitors fail to differentiate when populations are small is a known phenomenon for which there is a large body of literature. We have now included a brief summary of what is known in the Discussion section.

In addition we now include data showing that condensation fails to occur in the *Shh;Apaf1* DKO. In order to make it easier to appreciate the differences in sizes of the various simulation outcomes for each phenotype, we have included graphs of the turtle counts in Figure 6—figure supplement 1 for each of the 10 example runs used.

An unexplained aspect is that the Apaf1 somite is reduced in size by 25% around the time that Shh-mediated specification by Shh from the notochord and floor plate should be occurring. This would seem to predict that Shh could act on a greater portion of the somite and hence could specify a greater number of cells to a rib fate or lead to an increase in proximal rib fate as in Figure 6, yet this is not observed.

Thank you very much for pointing this out. Based on this comment, we have altered the NetLogo model so that the density is kept constant regardless of the initial size and hence smaller initial sized runs occupy a smaller area. Thus, starting with a smaller initial size, does lead to a higher percentage of cells subject to higher levels of the morphogen. We demonstrate this behavior in Figure 6—figure supplement 2 in which we provide example runs varying the parameters one at a time to see how they affect the results. In the case of the *Apaf1* KO, we found that the reduced initial size as estimated by the hierarchical Bayesian model, is not sufficient to have a strong impact on the overall pattern.

Another aspect that is not explained by the model is the spatial patterning of the two segments in a Shh KO or Shh;Apaf1 dKO wherein distal chondroprogenitor are seen distally but not proximally and in the Shh KO the distal cartilage is formed next to the sternum in the absence of proximal rib cells (or proximal chondroprogenitors).

The agent-based model addresses questions of early specification and the initial stages of outgrowth, however it is not intended to model the motion or outgrowth of the body wall and full closure of the rib cage. At E12.5, wild type animals do not have a fully enclosed rib cage and the body wall has not closed. The events leading to closure of the body wall and the mechanical movement of the rib cells into their final position is thus later in time and as such our model does not take extra efforts to model the movement of distal cells into their distal location. We think it is likely that the regions destined to be distal get loaded with cells first and then the distal skeletal compartment along with the other associated tissues expands and moves toward the midline.

If a preprint cannot be cited, the authors should provide arguments as to why the parameters and the structure of the model are biologically plausible. Methodological details of the model are insufficiently stated, and while the code is provided, this hinders reproducibility and assessment of model justification.

We now include a substantially more detailed description of the structure and justification for the model, as well as the choice of parameters used in Supplementary file 2 and Table 1, Figure 6—figure supplement 1, Figure 6—figure supplement 2, and 3. This material includes algorithm pseudocode, description of the cell motion model, and substantial analysis of the best fit parameters based on measured data from embryos.

Furthermore, it is not always clear in which way model outcomes and experiments were compared. For example, what constitutes "replicating the Apaf1 KO" given that this is a very small or no effect? If the point is simply that no change is observed when cell death and proliferation are lowered in concert, that is to be expected. It's fine to illustrate that with a simulation, but the point needs to made more clearly. Was the choice of parameter perturbations to replicate gene KO an informed one, or was the parameter space systematically searched to find "best matches?

Previously we were choosing parameters by determining if the visual outcome matched the phenotype. We have now refined our strategy using principled-choice of parameter values for the normal condition and then data-based choices for size and proliferation parameters as determined by our statistical model. We have explained the approach, which generates a more accurate and refined simulation, in the body of the paper as well as in the supplemental section (Supplementary file 2).

The manuscript could be strengthened by emphasizing the interplay between simulations and experiments more strongly and sooner. As it stands, there is a lot of "classic" experimental work to get through before getting to the simulations, and even then, the prediction of the simulations (about somite size and proliferation rate in the DKO) is not very clear in the figure.

We condensed the Results section and moved the figure on somite patterning to Supplemental (Figure 3—figure supplement 2) so that the reader can get to the model sooner. We adjusted the text, figures, and included a substantial accompanying Supplemental section (Supplementary file 2, Table 1) to help explain the method for choosing the parameter values and to better illustrate the outcomes.

Quantification of the Sox9 domain at E12.5 in whole mount was performed, but the data are not shown and the method is not described.

Our initial quantification was crude and so we simply gave an approximation. Upon further assessment, we decided that an accurate quantification was not really feasible as there was variability in how many rib aggregates were present and there was no way to identify which rib was which to make strict comparisons. In addition, the staining for the DKO was more diffuse compared to the SKO and it became difficult to be certain we were measuring accurately. We thus decided to omit the statement about quantification and simply state that there is a qualitative difference with the DKO ribs being smaller that is readily observed in Figure 4. We also now show cross-sections of these elements showing that condensations are not observed in the DKO situation.

What is the positive control for Apaf1 KO working, given the effect is hardly seen here?

All the animals were genotyped using standard PCR methods and this is now explicitly stated in the Materials and methods section. In addition, we could always identify the *Apaf1* KO animals because they have unusually shaped brains/heads due to overgrowth.

The method for quantifying the cross–sectional area is not described. How were the sections selected for analysis, what is the sample size for each genotype? pHH3 does not appear to label many cells in Figure 7, so the biological meaning of the changes are not clear. How many sections/embryo were analyzed? How many embryos were analyzed for each genotype?

All the animals were genotyped using standard PCR methods and this is now explicitly stated in the Materials and methods section. In addition, we could always identify the *Apaf1* KO animals because they have unusually shaped brains/heads due to overgrowth.

The manuscript could be shortened overall (especially the Introduction and Discussion section).

We made the Introduction, Results section and Discussion section more concise and trimmed the total text reducing it from 21+ pages to 19 pages.

[Editors’ notes: the authors’ response after being formally invited to submit a revised submission follows.]

Thank you for choosing to send your work entitled "A minimally sufficient model for rib proximal-distal patterning based on genetic analysis and agent-based simulations" for consideration at eLife. Your letter of appeal has been considered by a Senior Editor and a Reviewing editor, and we are prepared to consider a revised submission with no guarantees of acceptance. <<—This section seems repetitive as a similar statement is above, can it be removed?

We are grateful to the reviewers for their careful consideration of our study and excellent suggestions for improving the manuscript. After careful consideration, we saw three categories of comments. As we proposed, we now address them as follows:

The first category concerned the desire to understand why much smaller costal cartilage anlagen in the double mutants fail to mature. From our perspective, we were surprised that loss of *Apaf1* failed to rescue the *Shh* mutant and made the phenotype worse with smaller cartilage domains prior to differentiation.

We therefore focused our attention in this study on understanding this aspect. However, we now realize that we neglected to properly discuss the failure of a smaller cartilage anlage to differentiate, a common phenomenon in various mutational contexts but not necessarily broadly appreciated. In addition, while cartilage biologists are familiar with the requirement for dense culture conditions to obtain reliable cartilage differentiation, this may not be generally known. There is indeed an important body of literature that addresses the molecular mechanisms that drive cartilage differentiation and we recognize that we failed to discuss these studies sufficiently and relate them to our study.

To remedy this deficiency, we have completely re-arranged Figure 4 (previously Figure 5) to start with the early cartilage markers and end with the more differentiated markers. We also include a diagram with the steps of differentiation depicted in cartoon form so that it is easy to understand at which step the cells fail to move forward in double null animals. In addition, we have included 6 new panels showing that cells in the double mutant coalesce to form an aggregate but do not form a compact condensation.

In the Discussion section, we now include a summary of previous in vitro and in vivo studies concerning the molecular biology of differentiation (e.g. Retting et al., 2009; Barna and Niswander, 2007; Benazet and Zeller, 2012). Taken together these studies suggest that cartilage cells themselves produce factors including BMPs and TGFb at these pre- and early condensation stages which they then respond to by differentiating. Notably, in a study from Rolf Zeller’s lab, *Smad4* an intracellular signal transduction component required for responding to *both* BMP and TGFb signals was shown to be required *within* precartilage cells for differentiation to proceed. In the absence of *Smad4*, *Sox9*-expressing cartilage cells fail to compact, fail to undergo differentiation, and any condensations that form are not maintained. In addition, they provide results to support the idea that in the absence of sufficient cartilage differentiation signals, as would be expected when cartilage precursors are sparse, the cells adopt a connective tissue fate. We now also include reference to an interesting paper from Stuart Newman’s group in which a model is proposed that could be the sequel to our agent-based model (which stops prior to differentiation). They develop a discrete grid-based 2D multiscale computational model which addresses the process of cartilage condensation/differentiation, including consideration of cell number, cell behavior, and importantly the role of TGFb mediated cell-cell signaling in differentiation.

A second category concerned the description of the Agent-Based Model and some modifications, clarification, and statistical analysis of the model that needed to be included. We have addressed all the suggested changes. Importantly we now use the data we collected to improve the parameter estimates for the simulations. This resulted in a refined overall picture that is more accurate. In addition, we now include a multi-page supplemental section (Supplementary file 2) that describes the equations used, outlines the parameters with more information on the basis for choosing their values, and provides pseudocode similar to such sections found in two papers by McClennan et al., 2015. We now also include a supplemental section that shows the output for each mutant context in more detail as well several new figures that demonstrate other behavioral characteristics of the model.

The third category concerned problems surrounding data display, data presentation, and statistical testing. More specifically we generated more embryos to increase the N’s for the quantified samples, have included the raw data for the samples assayed for each set of experiments, and provided a substantial statistical analysis for quantification shown in Figure 5. We also addressed all data not shown problems, and reduced the length of the text.

We hope you will agree that an improved presentation of the results and a more in-depth discussion helps explain our phenotypes even to an audience that is not intimately familiar with the cartilage field. Thank you for your suggestions and detailed comments, these changes have greatly improved the manuscript and we hope you agree that it is now ready for publication.